# Beyond Land: A Review of Benchmarking Datasets, Algorithms, and Metrics for Visual-Based Ship Tracking

**Ranyeri do Lago Rocha \*** and **Felipe A. P. de Figueiredo**

National Institute of Telecommunication, 510 João de Camargo Avenue, Downtown, Santa Rita do Sapucaí 37540-000, MG, Brazil
**\*** Correspondence: ranyeri.rocha@inatel.br

**Abstract:** Object tracking has gained much interest in the last few years, especially in the context of multiple object tracking. Many datasets used for tracking provide video sequences of people and objects in very different contexts. Although it has been attracting much attention, no dataset or tracking algorithm has been applied to coastal surveillance and ship tracking. Besides image/video-based tracking technologies, other technologies, such as radar and automatic identification systems (AISs), are also used for this task, especially in maritime applications. In the AIS case, commonly known issues, such as information omission, remain to be dealt with. As for radars, the most important issue is the impossibility of identifying the ship type/class and correlating it with AIS information. However, image/video-based solutions can be combined with these technologies to mitigate or even solve these issues. This work aims to review the most recent datasets and state-of-the-art tracking algorithms (also known as trackers) for single or multiple objects tracking for objects in general and its possibilities for maritime scenarios. The goal is to gain insights for developing novel datasets; benchmarking metrics; and mainly, novel ship tracking algorithms.

**Keywords:** single object tracking; multiple object tracking; maritime surveillance; ship tracking; computer vision; machine learning





## 1. Introduction

Intelligent systems (ISs) are advanced machines that sense and react to the environment around them. They can assume many different forms, including autonomous vacuum machines, object tracking and identification, and facial recognition. This subject has gained a lot of attention in recent years, with certain tasks receiving greater focus in this research field, such as natural language [1], image, and video processing [2]. However, for the purpose of this review, we will specifically focus on image and video processing, which, as discussed throughout this article, has been used in various scenarios for detecting or recognizing pedestrians, cars, license plates, symbols, and objects in general. Despite the increased interest in IS for image and video processing, there is still a lack of research on datasets and algorithms specifically tailored for maritime surveillance tasks.

Several concepts and techniques from artificial intelligence (AI) are used to accomplish various IS tasks. Two main algorithms from AI that are widely used in IS are classification and regression. The first one aims to categorize (i.e., classify) objects into a discrete number of labels (known as classes), for example, cars, boats, motorcycles, and so on. On the other hand, regression algorithms aim to predict output values for specific input points of interest that have unknown outcomes.

In the classification field, we can distinguish two sub-fields: detection and recognition. Although they may seem similar, the two sub-fields have distinct characteristics. When detection is used, the system is generally not concerned with differentiating between different types of the same object. For example, in maritime scenarios, boat detection simply determines whether there is a boat in the scene. On the other hand, recognition

involves identifying the detected object and further classifying it into sub-classes. In the same maritime scenario, the system determines whether there is a boat and, if so, what type of boat it is. This concept is also frequently applied to face recognition.

The concepts described above collectively form what is known as computer vision (CV). The CV pipeline consists of image/video capture and processing, followed by detection, classification, and recognition tasks (see Figure 1). While earlier computer vision systems often required various image manipulation processes such as resizing, cropping, color space transformation, and filtering, nowadays, most computer vision benchmarks treat images with minimal pre-processing. The convolutional layers in deep neural networks extract features from the collected images. In many cases, resizing the image is the only necessary step. Thus, the journey described thus far supports the idea that tracking tasks are part of the CV field, and our review focuses on these tasks.

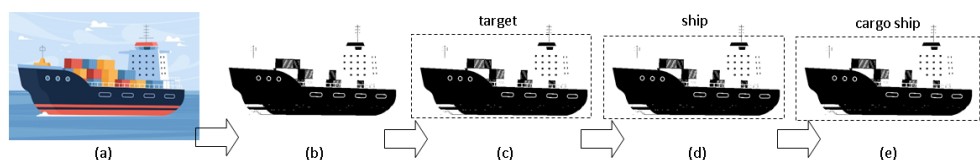

**Figure 1.** Pipeline for CV. (**a**) Collected image. (**b**) Processed image. (**c**) Detection. (**d**) Classification. (**e**) Recognition.

Object tracking has recently garnered significant attention due to the increasing computing power and the expanding number of applications, with new tasks emerging daily. In [2], the authors presented various application scenarios including traffic monitoring, robotics, autonomous vehicle tracking, and more. Additionally, in [3], the authors discussed applications for person-following robots, self-driving cars, and surveillance cameras.

The objective of the object tracking task is to establish the trajectory of an object between frames in a sequence [4]. This task can be divided into two primary sub-tasks: single object tracking (SOT) and multiple object tracking (MOT). In SOT tasks, the tracker aims to locate and track a single object in a sequence of frames to generate its trajectory. On the other hand, MOT tasks involve locating and tracking multiple objects of interest in frame sequences. Several benchmark datasets used for SOT, such as LaSOT [5], OTB-2015 [6], and VOT [7]. For MOT, datasets like the MOT series, including MOT15, MOT16, MOT17, and MOT20 [8–10], are commonly used.

Although there has been a recent surge in object tracking tasks and applications, there is still a significant research gap in the field of maritime surveillance. Specifically, there is a lack of benchmark datasets and algorithms used for visual ship tracking using modern approaches. Additionally, there is a shortage of standardized metrics that can be utilized to measure and compare the different algorithms. While several datasets used for neural network approaches in object detection, location, and classification tasks include ships, as reviewed in [11], the development of trackers or datasets specifically designed for maritime scenarios is yet to be seen.

Two important factors regarding the challenges in visual maritime object tracking can be highlighted:

1.  There is a lack of established benchmark datasets for ship detection and trajectory prediction. Many published studies rely on databases created by the authors solely for for their studies, with no publicly shared data available.
2.  There is a wide range of metrics defined and utilized in the literature, often leading to conflicting results and making fair comparisons between different tracking algorithms difficult. While [12] standardized some metrics for MOT tasks in their work, these metrics are not consistently employed. It is common to encounter variations of these metrics and the introduction of new ones for SOT or MOT. Essentially, many metrics assess the same information from different perspectives. Consequently, the definition of metrics specifically designed for maritime environments can be crucial.

In maritime surveillance tasks, two systems are widely utilized for data collection from ships: automatic identification systems (AISs) and radar systems operating at X-band frequencies. Ships equipped with AISs utilize a very-high-frequency (VHF) channel for communication, enabling the transmission of various tracking-related information, including maritime mobile service identities (MMSIs); static details such as International Maritime Organization (IMO) number, length, and ship type; dynamic data like GPS position, speed, acceleration, and tracking; and voyage-related information such as estimated time of arrival (ETA), cargo type, and passenger count. AIS receivers are capable of capturing and processing this ship and tracking information. Alternatively, maritime radar equipment operating in the X-band frequency range from 8 to 12 GHz can also be used to collect data. These systems employ a rotating antenna to emit microwave beams and receive the reflected waves using the same antenna.

Although there are established methods for obtaining tracking information, there are important limitations to their usage, as highlighted in [13]. The use of an AIS is not mandatory for all types of ships and voyages, and even when it is mandatory, there are no guarantees regarding the accuracy and completeness of the AIS information transmitted. Additionally, slow information updates can result in inaccurate tracking, especially for fast-moving ships. These issues can be addressed by employing a maritime radar system, which can quickly detect ships without relying on AIS data. However, despite the advantages of radar systems, they encounter challenges in detecting small targets and accurately identifying detected ships. To mitigate these problems, a combination of AIS and radar systems can be utilized.

With advancements in image processing techniques and computing power, as well as the growing need to address the challenges associated with AIS and maritime radar, visual ship tracking has emerged as a field of development. To provide an introduction and contextualize the content, we briefly list some work related to ships and maritime surveillance in visual tracking tasks, which can be found in the next section. The objective of this review is to shed light on how maritime object tracking has been utilized in recent years, specifically through the use of visual information. Furthermore, this review aims to provide a comprehensive description of benchmark tracking methods that are currently recognized, even if they were not initially developed for maritime tasks. Ultimately, this work will guide readers towards suitable resources for adapting or creating novel algorithms specifically tailored for maritime surveillance using tracking techniques.

The detection task plays a crucial role in tracking tasks. In this context, a tracker needs to detect an object and decide whether to track it frame by frame. In the maritime scenario, target objects can vary in size, which poses challenges for CV techniques to accurately process and detect them. Unfortunately, these techniques may struggle to distinguish or detect smaller ships. The literature has explored various approaches to address the specific challenge of small ship detection, including the use of convolutional networks [14], YOLO [15,16], and deep learning [17]. Research on small ship detection is of utmost importance in the context of maritime security, as imprecise detection of small targets can present significant challenges for ship systems. However, it is important to note that this review does not explicitly cover small ship detection. In terms of visual ship tracking, the presence of small ships can potentially impact tracking performance. However, this aspect will be considered for future analysis and exploration. It is worth mentioning that a similar exclusion applies to AIS and radar-based tracking methodologies.

This review aims to explore the treatment of ship tracking over time and to understand the functioning of systems, frame by frame, from simple tracking to the creation of tracking trajectories. It is evident that metrics and datasets are not standardized in maritime surveillance. Even in non-maritime scenarios with benchmark datasets for tracking, different metrics are defined, often specific to individual datasets, and sometimes, these metrics convey the same information. This makes it challenging to compare different studies effectively. By the end of this review, the reader should have a clear understanding of metrics used in tracking and the importance of datasets specific for tracker development. This re-

view also examines trackers used in other contexts, such as pedestrians, cars, motorcycles, and so on, with the aim of showcasing the most recent approaches to object tracking for single or multiple objects. Furthermore, throughout the review, we highlight the potential application of ideas from these works in the maritime scenario.

The work is organized as follows: in Section 2, we describe related work about visual maritime vessel tracking and the used tools and methods; in Section 3, we present the metrics commonly employed in most recent tracking systems for single or multiple objects; and in Section 4 we provide a detailed overview of various trackers grouped according to benchmark datasets. Section 6 offers a brief discussion on the metrics and datasets presented in this work, along with some recommendations for maritime tracking tasks. Finally, Section 7 presents the conclusion.

## 2. Related Work

This section describes some studies related to visual-based maritime tracking using different approaches. Some studies demonstrate the interactions between AIS and radar information, highlighting how visual tracking can enhance surveillance in maritime scenarios. Due to the scope of this review, only visual-based trackers are presented. It is worth noting, as aforementioned, that there are other tracking methods based on AIS and radar. References to studies involving AIS and radar (X-band and others) can be found in [18–24].

In [25], an automated intelligent video surveillance system for ships called AIVS3 was proposed. It aimed to capture, process, and generate tracks with detection, classification, and motion information in maritime scenes. The original work describes the algorithm flow with the following steps: (1) maritime target segmentation/detection; (2) classification of each target; (3) tracking of each target on each frame; (4) extraction of the location/position relative to the ship if necessary; and (5) and (6) implementation of an event/activity analysis engine to utilize the output of previous steps, such as tracks with set attributes, and mapping it to rules for maritime threat detection. These last steps are considered the core of the entire algorithm.

The results presented in the original demonstrate show a reasonable algorithm for target detection and trajectory construction, even in multi-target scenarios. However, the entire system was not compared with other algorithms. The authors only compared the classification step. In this work, several decision tree algorithms known as CART were utilized to form an ensemble arrangement called decision forest. Their proposed classification step was compared with Support Vector Machine (SVM) and the multi-class Naive Bayes classifier, showing the highest performance. No benchmark dataset or metrics were described to measure the tracking performance.

There are metrics used for potential target region detection and for constructing tracking trajectories during the execution of the tracker. The results demonstrate an excellent ability in terms of boat classification compared with SVM and Naive Bayes, achieving the same or higher scores in each categorical class. The figures in the original paper depict the results of the algorithm for target detection and trajectory construction. It is important to emphasize the proposed algorithm's robustness to occlusion.

The authors in [26] designed a technique for automatic detection and tracking tasks in open sea scenarios. An image sequence is collected and processed, involving edge and horizon detections, which are then used to predict and create tracks for the maritime ships. Horizon detection is widely employed in many studies for tracking maritime vehicles, and the authors describe this step as "the most important" in the algorithm. By determining the horizon parameter, the algorithm can discard edges that may not belong to a floating object. Two additional steps are performed: one for post-processing, which aims to create a preliminary segmentation of objects, and labeling regions, which output only regions adjacent or close to the horizon. In the end, the algorithm outputs the coordinates of the centroid, width, and height of the bounding box. The described steps are for a single frame.

The tracking task is performed after detection in the current frame, and a linear Kalman filter is used to predict the location and to estimate the bounding box. A track is

only created if the detected object appears in two consecutive frames and its bounding boxes intersect. As described by the authors, "new tracks are initiated for the objects that enter the frame or at the beginning of the video. The tracks are terminated when objects leave the frame".

Regarding the used dataset, the authors collected their own video sequences, and no benchmark dataset was used to evaluate the algorithm. Here, we have another work where no benchmark dataset is employed for performance evaluation in maritime scenarios.

The authors used three metrics that are based on Frame Detection Accuracy (FDA). They calculate the ratio between the overlap of bounding boxes and the mean value of the number of ground truths and detected objects in a specific frame. This metric can be applied to frame sequences, resulting in a metric called Sequence Frame Detection Accuracy (SFDA). Two variations were used: SFDA with a non-binary threshold and SFDA with a binary threshold.

It is important to note that in the next section, we will several metrics used in the most recent tracking algorithms. The bounding box overlap is included, but with different approaches and names. This highlights the importance of standardized metrics in tracking scenarios.

In another work by the same authors [27], an algorithm is proposed to operate in the open sea, away from any visible coastlines. As described in the paper, "this involves detection, localization, and tracking of ships", and the output consists of images showing the detected targets.

The first step is horizon detection. This process is crucial as the frame needs to have a clear distinction between sea and sky. Since the horizon line is a prominent feature in images of open seas images, only images with a high confidence in horizon line detection are considered. This aids in the subsequent step of image registration, which corrects and aligns the frames, simplifying the tracking process by using the horizon line as a reference.

The third step is segmentation, which localizes regions with ships. In this step, distractions in the image are discarded, and the focus is solely on potential targets. It is important to note that all targets are expected to be above the horizon line. The attentive reader should recognize the use of the horizon line in this step as a common approach seen in other studies presented in this section for the initial stages of ship tracking.

The final step is tracking, which is performed using the targets detected in the current frame and predictions made using the tracking parameters from the previous frame. The predictions are generated using a linear Kalman filter. The centroids and dimensions of the bounding boxes are used as filters to refine the track. If a new object is detected outside a valid region, a new track is initiated. Finally, a track consisting of more than ten frames is considered as a maritime ship.

In the original work, each step is illustrated with images to provide visual examples. The work incorporates several metrics at different stages, including metrics for horizon line detection. However, for the purpose of this work's scope, the specific metrics related to horizon line detection are not presented. On the other hand, metrics such as the number of detected objects, missed objects, false positives, and localization errors of detected objects are utilized for ship detection in individual frames.

The metrics utilized in this work adhere to the concepts outlined in [26]. The area of the bounding box intersection between the ground truth and detected objects determines the overlap. These two metrics are referred to as precision (PrecisionLoc) and recall (RecallLoc) of object localization. Precision is calculated as the ratio between the intersection and ground truth, while recall is calculated as the ratio between the intersection and the detected object. To address the asymmetry between recall and precision, the authors introduce the Dice metric, derived from these two metrics. By employing a predefined threshold in the Dice metric, the detected location is classified as an object, missed, or false alarm.

The precision, recall, and Dice metrics are extended to the sequence level to measure the overlap in the frame sequence. The final performance metrics used are Precision, Recall, PrecisionSequence, and RecallSequence, all defined under a pre-defined threshold. The

authors collected the dataset using the platform designed in [28]. This dataset consists of a large number of video sequences, unlike previous studies that only used a few sequences. For performance evaluation, 550 videos were utilized, each containing 100 frames, resulting in a total of 55,000 images. Other datasets were also employed to establish thresholds, to assess horizon presence, and to determine object localization. However, the authors do not compare the performance of the proposed algorithm with other approaches or systems.

In [29], the authors proposed an algorithm to stabilize ship images and to perform real-time tracking processing. It extracts parameters from the image and uses them to compensate for image non-stabilization. The compensation stabilization procedure is highlighted as one of the most important parts of this algorithm, as emphasized by the authors. "Even in better sea conditions, the ship's rolling, pitching, and yawing exist inevitably". In the introduction section of their work, the authors presented various factors that make video stabilization necessary, such as background, waves, reflection, and so on.

The algorithm's first step is the extraction of the horizontal line. It utilizes a multi-dimensional wavelet filter and a Hough transformation to obtain the parameters of the straight-line equation. The subsequent step is the video stabilization process, which involves applying a rotation matrix and a shift factor in different directions. However, it is worth noting that the shift factor, when combined with the rotation matrix, may result in blind areas where pixels have a value of 0 near the boundary. The authors provide an image to illustrate how compensation stabilization is performed.

After several computations, such as truncation center determination, horizontal line cut-off by the target, and estimation using a Kalman filter, the tracking of the target is performed. It is worth mentioning that this work also utilizes horizontal line detection, which is commonly employed in maritime tasks. Another significant aspect is that this work does not provide information about the dataset used, metrics employed, or performance data. In the conclusion, the authors mention that the algorithm was implemented on a patrol ship with "a strong capacity in real-time processing". However, without performance data and specific dataset specifications, it is challenging to grasp the true meaning of the term "strong capacity" as stated by the authors.

Another work [30] presented ARGOS, an algorithm for a surveillance system in the Venice waterway. It differs significantly from other algorithms described earlier in that the target detections are performed against a water background, which is a result of the camera's position.

As described previously, the other algorithms perform target tracking using the horizon line as a reference. In contrast, ARGOS performs optical detection and tracking of moving targets within each of the 14 cells defined inside the Grand Canal waterway. Information about the position and speed of each tracked boat is sent to the control center. A survey cell is a collection of cameras that together provide a field of view along the waterway.

The main steps include the local segmentation in the control center and the Multi-Object Tracking (MOT) module. The segmentation relies on an important feature called Optical Flow, which helps distinguish boats moving in different directions and removes noise caused by waves. However, it is not as effective when boats are very close to each other and moving in the same direction. The output of the segmentation serves as input to the tracking module, which is responsible for generating trajectory data. The tracking module employs a multi-hypothesis tracker based on a set of Kalman filters.

That study evaluated the algorithm using three metrics in various weather conditions and dates: false negatives (boats not tracked), false positives due to reflections, and false positives due to waves. The weather conditions considered were cloudy/foggy, sunny/cloudy, cloudy, cloudy/rainy, and sunny. The algorithm's performance was found to be the poorest in the most adverse weather conditions. Additionally, the authors conducted tests to count the passing boats, measuring false negatives and false positives. It is worth noting that ARGOS does not rely on a specific dataset since it was designed to operate continuously in real-time, 7 days a week, 24 h a day.

The authors in [31] presented an approach for detecting and tracking ship targets in a cluttered forward-looking infrared image sequence. The work describes clutter scenarios affected by atmospheric radiation, the sun's refraction, bright clouds, sea clutter, and imaging distance. The term FLIR is used for infrared images, which stands for forward-looking infrared (FLIR).

That paper presented a system for real-time detection and tracking using important steps that is described next. First, the Target and Background Modeling step occurs, which aims to separate the background, the dark sea surface, from the target, the brighter part on infrared images. This part aims to model the attributes of noise, target, background gray value, and size. The second part corresponds to ship target detection, which involves image preprocessing with a Gaussian low filter to remove noise and to enhance the image quality. In the sequence, a multilevel filter is designed to extract the target from the image.

The authors explain that the target size ranges from $1 \times 1$ to $7 \times 7$, and the image scaling operation is performed if it is exceeds this range. To perform this step, many parameters are defined to facilitate the scaling operation. In the next step, the image is passed through a multilevel filter, where different filters are used to suppress the background and to enhance the target. As described in the original work, "the target is brighter than the background", in general. Once the target is identified in the image, several segmentation processes are carried out, using OTSU's method to determine the threshold for binarization and the localization of the sea–sky region.

The last important step is the ship target tracking. The authors utilize the characteristic of infrared images with a distant target to implement an algorithm for detecting objects in complex scenarios. This step employs an algorithm for vision processing called histogram back-projection, in which, given a region of interest (ROI), the algorithm searches for regions in the original image with a similar histogram shape. The mean-shift tracking algorithm is used to track the target, and at the end, one step is performed for target verification and localization to ensure accurate identification of the target candidate.

This work utilized a system to collect videos, each comprising 500 frames. The image sequence was captured using an infrared system. The images were manually labeled and stored as a ground truth database, classified into low- and high-clutter scenes. The dataset is not publicly available, and the authors do not provide any information on how to access the image sequence for reproduction. The target size used for performance evaluations was a $5 \times 5$ pixel rectangle corresponding to each pair of ground truth target coordinates.

The system outputs a rectangular block with a size of $5 \times 5$, which is compared with the ground truth target. The algorithm utilizes a simple intersection to determine the correct classification. If multiple blocks intersect with the target ground truth, they are all classified as a single target. Additionally, the false alarm count is calculated by subtracting the number of objects classified as targets from the total number of detected objects.

For comparison, the authors utilized the corrected detection rate, which represents the rate of system output that is classified as a true target and is indeed a true target. Another metric used is the false alarm rate, which indicates the rate of system output that is classified as a true target but is, in fact, a cluttered scene. A receiver operating characteristic (ROC) curve is plotted using these two metrics. In this work, the proposed method is compared with two different algorithms, namely Zhang's method [32] and Braga-Neto's method [33].

These methods have certain deficiencies compared with the one proposed in that work, particularly in background analysis and target size. The proposed method achieved a success rate of 90%, while Zhang's method scored 76% and Braga-Neto's method scored 68%. Additionally, the proposed method exhibited a shorter time consumption for target detection, taking only 0.8 s compared with 1.1 seconds for Zhang's method and 1.3 s for Braga-Neto's method. These results were obtained in highly cluttered images.

In [34], the authors proposed a tracking algorithm for infrared images with the sea-surface target. They developed a tracking algorithm using two computer vision algorithms: Scale-Invariant Feature Transform (SIFT) and Bag of Features (BOG) for detection and tracking tasks. Both algorithms were presented in detail in the original work.

The proposed target detection and tracking algorithm includes a training phase in which features are learned for BOF. As described in that work, "this phase can be summarized as the clustering of SIFT descriptors extracted from the target and background regions". Several tasks are performed: horizon detection based on the intensity level of the sky and sea; target/background determination, where the operator specifies the target with a bounding box; and the SIFT descriptors, used to describe the target within the bounding box. Finally, the algorithm utilizes K-means clustering to cluster the target and background SIFT descriptors. The centroids produced by the K-means algorithm are referred to as "visterms", according to the authors.

After the training phase, the test phase is conducted, primarily focusing on target detection and tracking. There are three main steps in this phase: descriptor classification, where the proposed algorithm is responsible for classifying the SIFT descriptors as either the target or background; descriptor matching, where a suitable algorithm is used to match the SIFT descriptors in two consecutive frames, establishing a "link" for tracking task; and finally, region-of-interest (ROI) selection. This step determines the minimum rectangle that encompasses all the matched SIFT descriptors. For detailed information about each step in the training and test phases, please refer to Section 4 in [34].

To evaluate the algorithm, several metrics are employed to compare the proposed tracker with the Kanade–Lucas–Tomasi (KLT) feature tracker method [35]. The metrics used are as follows: (1) Euclidean distance between the centers of the ground truth target region and the detected target region.; (2) city block distance between target centers, as described in the previous metrics; (3) the ratio between the "under-detected target area and the total target area", referred to as the false-negative rate. This metric indicates the proportion of the target area that is not covered by the detected target area; and (4) the true-positive rate, which calculates the ratio between the "correctly detected target area and the total detected target area". This metric represents the proportion of the target area that is covered by the detected target area. It is important to note that all metrics involve distance measurements between the ground truth and the detected target.

This work utilized three datasets to evaluate the performance of the tracker: real infrared videos, synthetic infrared videos, and visual band videos. All videos were either recorded or generated; however, there is no information available on how to obtain the samples used in this work. For the real infrared videos, a total of 11 videos were employed, with three videos designated for training and eight videos designated for testing. The infrared videos used in the evaluation were specifically acquired for this purpose, following the setup described by the authors. The videos were captured in different locations, in different orientations, and at various times of the day. As for the synthetic videos and visual (VIS) band videos, only the frame sizes were provided; no additional details were given. The authors presented the complete results in the original work, including tables displaying the metrics' outcomes for each dataset. In general, the KLT method faces challenges with illumination changes, and in most cases, the proposed method exhibits superior performance.

It is important to note that eleven video sequences were utilized, encompassing various scenarios, orientations, and times of day, among other factors. Despite these differences, the proposed method demonstrated robustness in performing detection and tracking tasks.

The authors proposed a ship tracking framework for tracking and ensuring ship position accuracy in their work [36]. The framework utilizes the kernelized correlation filter (KCF model) via curve-fitting, referred to as KCFC, which incorporates the KC model for ship tracking and a curve fitting algorithm for outlier removal.

The first step involves implementing the KCF model as a KCF ship tracker by correlating the ground truth ship sample with the candidate sample. The ground truth ship sample is manually labeled, and the trained KCF tracker distinguishes the target ship by evaluating the maximum stimulus response between the ground truth sample in the previous frame and the ship's current position in the current frame. This process determines the potential ship position in each image. For achieving the maximum response, ship candidates, impor-

tant training data, and operators are transformed into the frequency domain using discrete Fourier transform. The second step focuses on addressing a limitation of the KCF model, which is its susceptibility to obstacles moving in neighboring water channels. To overcome this issue, the trained KCF model is enhanced to extract ship features from neighboring ships instead of mistakenly identifying them as the target ship. In order to mitigate this problem, an outlier denoising procedure is implemented. Initially, outliers are detected, and subsequently, a curve fitting model is employed to reconstruct the ship trajectory in the occlusion image. The authors emphasize that the curve fitting model assumes that ship motions during occlusion remain consistent, meaning that the motions do not undergo significant changes.

To quantify the performance of the ship tracker, four statistical metrics are employed: root mean square error (RMSE), root mean square percentage error (RMSPE), mean absolute deviation (MAD), and mean absolute percentage error (MAPE). The authors assess the tracker performance by measuring the Euclidean distance between the ground truth and tracked center points.

The dataset used consists of four videos collected by the authors, with the duration ranging from 21 to 37 s. These videos include various occlusion scenarios, such as a small-size ship occluded by a large-size ship or a large-size ship occluded by a large-size ship (either completely or partially). It is important to note that the dataset is not publicly available and is not considered a benchmark.

The KCFC tracker was compared with other models, such as KCF (pure), mean-shift, and the Scale Adaptive with Multiple Features tracker (SAMF). The results demonstrate that KCFC can accurately track the position of the ship even in occluded scenarios, whereas other models fail to do so. The original work provides detailed information about the tracking results for the four videos.

The work proposed in [37] introduces a simple yet efficient method for tracking and analyzing the movement direction of the ships. The proposed tracker utilizes video processing techniques, starting with frame difference to distinguish the foreground from the background. It incorporates various image processing operations, including denoising, erosion, dilation, and connected component labeling, to isolate the moving object within the image. The moving object, referred to as the ship in this context, is defined using a bounding box.

Going deeper into the video processing technique used in [37], the algorithm employs a reference image from the background, which is then subtracted from the current frame image. The resulting difference is transformed into a binary image with black and white pixels. To reduce white noise pixels, a median filter is applied to the binary image. Additionally, erosion and dilation operations are performed to address any holes that may have been generated during the ship segmentation process. Finally, the Connected Components Labeling method is utilized to extract all the ships and to assign them labels.

To track the ships once they have been segmented, this work computes the foreground center point and uses this point to compare with other frames. The tracker defines a center point's coordinates as $X_i$ and $Y_i$, and the center point of the last image as $X_{i-1}$ and $Y_{i-1}$. The standard deviation is computed between these two center points.

This tracker was evaluated without the use of a dedicated dataset. The authors employed a camera positioned at the seaport and tested the tracker on the captured video. The experiments encompassed three distinct ship flow environments, ranging from a single flow to interlace ship flows. The sole metric employed for evaluation was accuracy, measured as the percentage of frames successfully tracked.

In another straightforward tracking algorithm proposed by the authors of of [38], they adopted the same fundamental idea as in [37]. The algorithm follows the following steps: collecting images from a stationary camera to create a reliable background, performing image subtraction on each subsequent frame to generate a foreground by calculating their difference. The resulting image is then converted into a binary representation, and morphological operations are applied to diminish noise and to fill gaps in the foreground image.

The target location method employed in this approach involves the utilization of a predefined filter value. This filter value serves to clean up the image both inside and outside, thereby forming a silhouette. If the size of the object is smaller than the filter value, it is discarded. A bounding box is then defined around the object, and a centroid, referred to as a central point, is computed. In contrast to the method proposed in [37], the authors in this study introduced the concept of inter-frame correlation to determine a correlation coefficient when the target moves in or out of the image. However, the details of this step are not explicitly described in that paper. It mentions the use of "tracking techniques" to consolidate the results obtained from tracking the moving target.

No benchmark dataset was utilized; instead, a video consisting of 770 frames was collected. The initial 30 frames, which did not contain the ship, were employed to estimate the background, while the subsequent frames were used to detect and track the moving object, namely the ship. However, the paper does not provide any defined or described metrics to measure the efficiency of the tracking process.

Another important aspect of ship tracking is the support it provides to traditional maritime tracking methods, such as X-band radar and automatic identification systems (AISs). AIS is utilized in maritime surveillance to supply tracking information. However, the information provided by AIS messages is only mandatory for certain types of boats, and even when it is mandatory, the information can be inaccurate or the AIS can be deactivated. On the other hand, the X-band radar is another widely used instrument in maritime surveillance scenarios. It can address the limitations of AIS as it does not rely on any information transmitted by a ship; instead, the radar system itself performs the tracking. However, despite the advantage of self-tracking, radar can be imprecise for small targets, and in some cases, it may not detect the ship at all. In a study conducted by Zardoua et al. [13], the concepts and limitations of AIS and radar are explored and compared with visual tracking. The authors emphasize the significance of employing camera surveillance systems in combination with AIS and X-band radar, as this approach overcomes the aforementioned issues.

In order to address these challenges, the authors of [39] employed visual active cameras to complement radar information and to aid in ship detecting and tracking in harbors. The proposed system utilizes a pan–tilt–zoom camera to track a specific ship. To initiate the process and to establish the feature point tracker, object detection is employed. The system utilizes a Histogram of Oriented Gradients (HOG) detector, which is specifically trained to detect the cabin of the ships. This detector is combined with the KLT feature point tracker, which utilizes small features within the object to continuously track it until the tracking is lost.

The utilization of active cameras may lead to feature points becoming disconnected from the object. To address this issue, a validation step is proposed in order to remove "problematic feature points" by considering the median position and displacement of each feature point. Additionally, to counter the decrease in the number of feature points, a re-initialization step is implemented using the HOG detector. Since the system employs an active camera, the subsequent steps involve calculating the control parameters for pan, tilt, and zoom adjustments of the camera.

This work was implemented in the port of Rotterdam, where the local vessel traffic system (VTS) was utilized. The VTS system was originally intended for radar data. To estimate the position of the ship, the pan, tilt, and zoom values are employed to calculate the distance.

No information regarding the dataset was provided in this work. The authors mentioned that the system tracked ships for a duration of 18 min and over a distance of 1.5 km. However, no specific metrics were presented in the original text, and there is no information about reproducibility.

Another work proposed a framework that combines AIS and visual tracking. In [40], the authors utilized AIS to enhance visual tracking by leveraging the vessel's positions obtained from AIS to dynamically adjust the camera's position, ensuring a focused view of

the target. The paper introduces two steps: adjusting the camera's pose and focal length and utilizing a Kalman filter to achieve smooth and gradual adjustments.

The AIS positioning information includes various fields, such as the ship's name, MMSI, position, time, etc., which are collected and stored in a database. These fields serve as input to correct the vessel's position. The AIS track position, specified by latitude and longitude, is adjusted to obtain the ship's center position. Similarly, the AIS fields are utilized to calculate the distance and the horizontal azimuth from the camera to the target. The coordinate values in the spherical coordinate system are determined using the Mercator Sailing Method, although the complete procedure is beyond the scope of this discussion. For further details, please refer to [40].

The second step involves applying a Kalman filter to achieve smooth vessel tracking. The Kalman filter is utilized to estimate the trajectory and to mitigate noise and water influence on the camera's adjustment. The complete kinematic model is detailed in Section 3.1 of the original work. To create a continuous AIS position track, the authors employed cubic spline interpolation. This interpolation technique is necessary because AIS messages are typically updated within a range of 2 to 5 s.

The proposed method was implemented in a simulated experimental system using Matlab. However, the authors did not provide any information about the specific AIS data used or the metrics employed to evaluate their work. The estimated camera parameters are presented, but no actual visual tracking was performed in real-world scenario.

In [41], the authors presented a method for detecting gaps in AIS information, which is not directly related to ship tracking like the previously described studies. However, we include it here to highlight the importance of addressing AIS anomalies in maritime surveillance. The authors emphasized the issue of transmission gaps in AIS data, which can arise due to factors such as poor transmission quality, disabled transmitters, high-density boat locations, and incorrect information. Unlike previous work, this method does not involve the use of cameras to support AIS information. Instead, a probabilistic model was developed to assess the likelihood of boats based on their AIS patterns. The model was tested using AIS data from the Arafura Sea, which was provided by the Australian Maritime Safety Authority and is not publicly available for download. The results demonstrated that the proposed model could effectively identify abnormal gaps in AIS transmission for two case studies with high-risk gaps. It is important to note that this work did not compare its performance with other existing methods or algorithms. The metrics used in this system differ from those employed in ship tracking and, as such, is not considered in this review. Consequently, this proposed method is not included in Table 1.

**Table 1.** Comparison about dataset, metrics, and performance evaluation.

| Work | Dataset | Metrics | Comparison |
|------|---------|---------|------------|
| [25] | No description | No description | Classification step compared with SVM and multi-class Naive Bayes classifier |
| [26] | Collected by the authors | FDA, SFDA NBT, and SFDA BT | No comparison was described |
| [27] | Collected by the authors | Precision, Recall, Dice, PrecisionSequence, and RecallSequence | No comparison was described |
| [29] | No description | No description | No comparison was described |
| [30] | Real-Time Data | False negative, false positive due to reflections, and false positive due to waves | No comparison was described |
| [31] | Collected by the authors | Correct detection rate and false alarm rate | Compared with [32,33]. |
| [34] | No benchmark | Euclidean and city block distance, false-negative rate, and true-positive rate | Compared with KLT [35] |
| [36] | Collected by the authors | RMSE, RMSPE, MAD, and MAPE | No comparison was described |
| [37] | Real-Time video | Frames tracked accuracy | No comparison was described |
| [38] | Collected by the authors | No metrics were described | No comparison was described |

In [42], a survey was presented on the use of deep learning algorithms for autonomous surface vehicles (ASVs). However, this topic differs from the main purpose of the present work, which focuses on visual approaches for ship tracking. ASVs have the capability to support coastline surveillance and to enable vehicle cooperation, but they require a maritime vehicle to perform these tasks. Therefore, while ASVs are relevant in the broader context of maritime surveillance, this review specifically concentrates on visual methods for ship tracking.

To summarize this section, Table 1 compares all trackers described based on the most important factors discussed: datasets, metrics, and performance evaluation.

This section is crucial as it provides an overview of the primary methodologies employed in visual maritime surveillance. However, it is worth noting that the current literature seems to overlook newer types of trackers that capitalize on the advancements in AI and CV algorithms. While Section 4 presents a range of trackers that employ the methods discussed here for diverse scenarios such as pedestrians, cars, and objects, the area of maritime surveillance using ship trackers remains an unexplored field. For visual ship detection, we can find recent techniques in [11,43] that utilize state-of-the-art approaches.

## 3. Review of Performance Metrics

This section describes the most known and used metrics for measuring and comparing the performance of different trackers in different datasets available in the literature. These metrics are used in the next section to compare the performance of algorithms using benchmark datasets. Additionally, it is important to note that there are no standardized, i.e., benchmarking, metrics used for judging different trackers and datasets.

### 3.1. Precision (P)

Precision is computed by comparing the distance, in pixels, from the center to the center, between the tracking result and the ground truth bounding box [44]. This metric can be used to sort different trackers using a threshold, for instance, 20 pixels. This metric is defined as

$$P = \|C^{gt} - C^{tr}\|_2, \tag{1}$$

where $C^{gt}$ represents the ground truth bounding box center and $C^{tr}$ represents the tracker bounding box center.

### 3.2. Normalized Precision (P_norm)

The authors in [44] proposed the normalized precision metric to address the sensitivity of precision to image resolution and bounding box size. This metric is defined as

$$P = \|W(C^{gt} - C^{tr})\|_2, W = diag(BB_x^{gt}, BB_y^{gt}) \tag{2}$$

where $BB_x^{gt}$ represents the number of pixels along the $x$ axis in the ground truth bounding box, and $BB_y^{gt}$ represents the number of pixels along the $y$ axis in the ground truth bounding box.

### 3.3. Area Under Curve (AUC) or Success

The AUC is commonly used for comparing classification models. In the context of tracking, the AUC is computed by plotting the normalized precision on the y-axis and the normalized distance error threshold on the x-axis. A 2D plot curve is defined with the axis being the false-positive rate (FPR) and the true-positive rate (TPR). The resulting curve represents the performance of the trackers, and the AUC is calculated as the area under this curve, ranging from 0 to 1. In the tracking context, the AUC is computed under the curve of normalized precision on the y-axis and the normalized distance error threshold on the x-axis. Another metric that can be used is the success score plot. Trackers are ranked based on the AUC values. In a study by Vcehovin et al. [45], it is shown that the AUC is

equivalent to the average overlap metric, which will be further discussed as the expected average overlap metric, referred to as S-AUC.

### 3.4. Accuracy (A ↑)

Accuracy is a measure of the overlap between the ground truth bounding box and the predicted bounding box. It indicates how well the predicted bounding box aligns with the ground truth. High accuracy is achieved when two bounding boxes perfectly overlap, meaning they have the same position.

### 3.5. Robustness (R ↓)

Robustness measures the frequency of tracker failures in detecting the target during tracking. A failure is recorded when the overlap between bounding boxes, as described earlier, equals zero.

The aforementioned metrics are summarized in Figure 2. As mentioned in the Introduction, multiple metrics have been developed to convey the same information. For example, Accuracy and Precision provide identical information, which is the distance between the ground truth bounding box and tracking bounding box. As we see in the following sections, various other metrics are derived from these and employed to assess the performance of trackers.

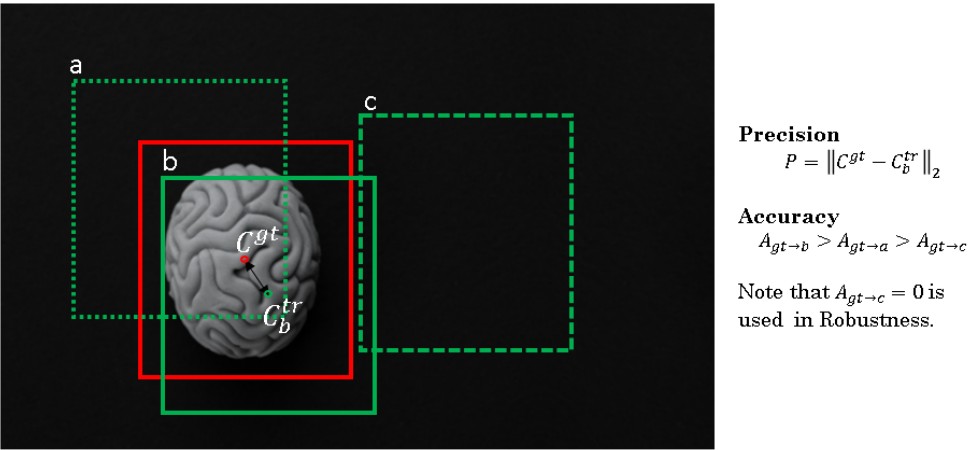

**Figure 2.** Precision, accuracy, and robustness concepts. The red bounding box is the ground truth. The letters a, b, and c are bounding box references in the image.

### 3.6. Expected Average Overlap (EAO ↑)

The authors in [46] mentioned that accuracy and robustness were utilized in VOT2013 and VOT2014 to rank tracking methods, but the results did not have a concrete application on tracking interpretation. To address this issue, VOT2015 introduced the expected average overlap (EAO). EAO calculates the average overlap between the predicted bounding box and the ground truth bounding box over time. It combines accuracy and failures in long sequence frames, denoted as $N_s$. The overlap at frame $i$ is represented as $\phi_i$. For a sequence with $N_s$ frames, the average overlap, including failures (zero overlaps), is defined as $\Phi_{Ns}$:

$$\Phi_{Ns} = \frac{1}{Ns} \sum_{n=1}^{Ns} \phi_i. \tag{3}$$

The expected average overlap is computed as the average among all sequences in the dataset. Each sequence has its own $\Phi_{Ns}$, and the resulting EAO is the average of the average overlaps, denoted as $\hat{\Phi}_{Ns}$. The next step involves plotting the curve of this measure for each value of $Ns$, ranging from $Ns = 1 : N_{max}$. This curve, known as the expected average overlap curve, is used to rank the trackers. Equation (4) shows how the final EAO value is computed over the interval $[N_{lo}, N_{hi}]$. The initial and final frame values are determined

using Kernel Density Estimation (KDE) calculated over the dataset sequence. When the PDF within the range is equal to 0.5, the range is fixed. Therefore, for each dataset, the values of $N_{lo}$ and $N_{hi}$ are fixed and can be used to compute the final expected average overlap measure, $\hat{\Phi}$.

$$\hat{\Phi} = \frac{1}{N_{hi} - N_{lo}} \sum_{Ns=N_{lo}}^{N_{hi}} \hat{\Phi}_{Ns} \tag{4}$$

A higher EAO value indicates a more accurate tracking algorithm. EAO is often employed in the context of single-object tracking, aiming to accurately track a single object over time.

### 3.7. Multiple Object Tracking Accuracy (MOTA ↑)

MOTA is a metric used to evaluate the performance of multiple object tracking algorithms. It quantifies the percentage of errors made by the tracker, encompassing false-positive and false-negative detections, missed targets, and identity switches. A higher MOTA value signifies a superior tracking performance of the algorithm. MOTA is extensively employed in object tracking research, especially for assessing the accuracy of tracking in intricate scenarios involving multiple objects, occlusions, and other challenging conditions. It is defined as follows:

$$MOTA = 1 - \frac{\sum_t m_t + fp_t + mme_t}{\sum_t m_t} \tag{5}$$

where $m_t$ represents the number of tracking misses or false negatives, $fp_t$ denotes the false positives, and $mme_t$ corresponds to the number of mismatching or identity switches at time $t$.

### 3.8. IDF1 Metric

The Identifier F1 score (IDF1) is a performance metric used to assess the accuracy of multiple object tracking algorithms. It quantifies the F1 score of the matching process between the ground truth and the predicted trajectories. The matching process involves correctly identifying the track IDs for the objects across video frames. The IDF1 metric considers both the precision and recall of the matching and is defined as the harmonic mean of these two measures. A higher IDF1 score indicates a more accurate tracking algorithm. IDF1 is commonly employed in the multi-object tracking scenarios, where the objective is to accurately track multiple objects over time. It differs from *MOTA*, which focuses solely on the detection level.

### 3.9. Higher-Order Tracking Accuracy (HOTA)

This metric was developed to provide a comprehensive evaluation metric for trackers that incorporates various evaluation aspects in a fair manner. HOTA extends traditional tracking metrics, such as MOTA and IDF1, by incorporating the spatial and temporal relationships between the ground truth and predicted bounding boxes. Additionally, HOTA introduces the concept of higher-order associations, which assess the relationship between sets of predicted and ground truth objects beyond pairwise associations. The HOTA score ranges between 0 (indicating poor performance) and 1 (representing perfect performance). HOTA has demonstrated increased robustness to tracking challenges, such as occlusion and clutter when compared with traditional tracking metrics. As a result, it is increasingly being adopted as a standard evaluation metric for object tracking algorithms. The formal definition of HOTA, along with other related metrics like $DetA$ and $AssA$, can be found in [47].

*3.10. Identity Switching (ID$_S$ or IDSW)*

Identity switching, also referred to as ID switches or ID swaps, is a prevalent error that can occur in multiple object tracking systems. It occurs when the identity of a tracked object is mistakenly swapped with another object's identity, often due to their similar appearance or behavior. For instance, if two objects are in close proximity and moving together, a tracking algorithm may erroneously interchange their identities, resulting in incorrect tracking outcomes. Identity switching can have a significantly impact on the accuracy of a tracking system, particularly in cluttered or complex scenes where distinguishing between objects becomes more challenging.

*3.11. Tracker Velocity (FPS)*

Tracker velocity refers to the speed and direction at which a tracked object is moving, as estimated by a tracking algorithm. The estimation of velocity is typically based on analyzing the positional changes in an object over time. However, velocity estimation can be affected by factors, including measurement noise, occlusions, and interactions with other objects. Different tracking algorithms employ different techniques for estimating object velocities, such as Kalman filters, particle filters, or optical flow analysis. In some studies, velocity is measured in Hertz. It is worth noting that while this metric is presented here, comparing the values obtained using this metric with those from other trackers can be challenging. This is because velocity estimation can depend on various hardware and software factors that may differ between different tracking systems.

It is important to acknowledge that using different metrics for different datasets can hinder the comparison of different trackers. While the metrics may be consistent within a specific dataset, this work highlights the possibility of unifying metrics even across different datasets. By establishing a common set of metrics, it becomes easier to compare the performance of different trackers and to draw meaningful conclusions.

## 4. Tracking Review

This section provides a description of four datasets that are widely used in various visual tracking scenarios, encompassing different types of objects. Additionally, it offers a review of several state-of-art trackers that can be utilized for visual ship tracking. While these trackers have been primarily employed in diverse contexts, they can also be applied to ship tracking tasks. Lastly, a performance comparison of the trackers is presented for each dataset.

*4.1. Datasets*

4.1.1. LaSOT

Large-scale Single Object Tracking (LaSOT) is a dedicated benchmark dataset specifically designed for training and evaluating single-object tracking (SOT) algorithms in video sequences [5]. This dataset follows several key principles, including large-scale coverage, high-quality bounding box annotations, long-term tracking scenarios, balanced class distributions, and comprehensive labels. With a total of 3.5 million frames, LaSOT is one of the largest tracking datasets available, featuring 70 different object classes across 1400 video sequences. On average, each video sequence in LaSOT consists of 2515 frames. In order to provide a comprehensive description, each frame sequence in the dataset has been labeled with 14 attributes, including Illumination Change, Full occlusion, Partial Occlusion, Target Deformation, Target Region Blurred, Fast Movement, Scale Changing, Rotation, Background Cluttered, Low resolution, Point of View Change, Out of View, and Bounding Box Ratio out of Range. For more detailed information on the sequence distributions for each attribute, refer to [5].

4.1.2. OTB-2015

OTB-2015 (Object Tracking Benchmark 2015) is another benchmark dataset widely used for visual object tracking evaluations. It comprises 100 video sequences of varying

lengths and difficulty levels, amounting to over 11,000 frames in total. The dataset encompasses a diverse set of challenging scenarios, including occlusion, motion blur, illumination changes, deformation, fast motion, in-plane rotation, out-of-view, background clutter, low resolution, and scale variations. The tracking performance on this dataset is assessed using various metrics, such as precision, success rate, and AUC. For a comprehensive description of the video sequences, including labeling distribution, refer to [6].

It is worth mentioning that many labels used in OTB-2015 are also employed in LaSOT. A comparative analysis of label distribution between the datasets is presented in [5]. Each sequence in both dataset includes a target object indicated by a bounding box. In general, LaSOT is regarded as a more comprehensive and challenging benchmark dataset compared with OTB-2015.

### 4.1.3. VOT-2016

The VOT-2016 dataset is a component of the Visual Object Tracking (VOT) challenge [7], an annual competition that strives to advance the state-of-the-art in visual tracking algorithms. It comprises 60 video sequences that encompass a diverse set of challenges in visual tracking, including scale variation, occlusion, motion blur, motion change, camera motion, illumination changes, background clutter, and unlabeled sequences.

Each video sequence in the VOT-2016 dataset contains annotations in the form of bounding boxes around the target object, which is the object of interest for tracking. Additionally, the dataset provides annotations for various challenges, including occlusion, out-of-view, deformation, and motion blur. These annotations serve as valuable resources for researchers to evaluate the performance of their tracking algorithms in diverse scenarios and to facilitate meaningful comparisons with other state-of-the-art trackers.

In the 2016 challenge, an automated technique was utilized to assign target bounding boxes. This technique involved a segmentation step aimed to separate the target object in each frame, followed by the definition of the bounding box through the optimization of a cost function.

### 4.1.4. MOT17

The MOT17 dataset is a widely used benchmark dataset for evaluating multiple object tracking algorithms, introduced as part of the Multiple Object Tracking Challenge in 2017 (MOT17) [9]. It is an extended and improved version of the MOT16 dataset and has gained popularity as a reliable dataset for benchmarking multi-object tracking algorithms.

The MOT17 dataset comprises 14 video sequences captured in highly dense environments, encompassing a wide range of scenarios. These scenarios include indoor and outdoor scenes; various camera viewpoints; and different levels of occlusion, crowd density, and object appearance. The dataset contains video sequences with a significant number of people, as well as other important object classes such as Pedestrian, Person on Vehicle, Car, Bicycle, Motorbike, Non-Motorized Vehicle, Static Person, Distractor, Occluder, Occluder on the ground, Occluder full, and reflection. The sequences focus on tracking moving objects as the primary targets.

The object classes in the MOT17 dataset are categorized into three groups: Pedestrians, which encompass both moving and stationary individuals; People, which include non-standing individuals and artificial representations of humans; and Vehicles and Occluders. The videos in the dataset were captured using multiple cameras, each with different resolutions and frame rates.

Each video sequence in the dataset is annotated with ground truth trajectories for all objects that appear in the video. The ground truth annotations include bounding boxes, unique IDs for each object, and additional information about occlusions and false detections. These annotations allow researchers to evaluate the performance of their tracking algorithms in diverse scenarios and to make comparisons with other state-of-the-art trackers.

The sequences in MOT17 exhibit significant variation due to the use of static and moving cameras, diverse weather conditions, and different points of view. In addition, MOT17 incorporates two additional detectors, namely DPM (deformable part model), Fast R-CNN, and SDP (scale-dependent pooling), for object detection in each sequence. Overall, the MOT17 dataset consists of three versions of the same 14 sequences, resulting in a total of 42 sequences.

### 4.2. *Trackers*

The studies discussed in this section represent the state-of-the-art in visual object tracking and make significant contributions to the field. However, it is important to note that each method has its own strengths and weaknesses, and determining the optimal approach depends on the specific application and tracking requirements. The research in visual object tracking is a rapidly evolving and dynamic field, with new work continually emerging, each offering its distinct approach, strengths, and limitations.

### 4.2.1. OSTrack-Joint Feature Learning and Relation Modeling for Tracking: A One-Stream Framework

In [48], a unified one-stage, one-stream tracking approach is introduced, with a focus on joint feature extraction and relation modeling. This approach differs from the commonly used two-stage, two-stream methods, which involve feature extraction in both the template and search regions. The joint approach proves to be particularly effective in tracking targets that undergo continuous changes.

The approach described in the work takes two input images: a template and a search region. It utilizes self-attention layers to establish a strong correlation between the target (represented by the template image) and each candidate (derived from the search region). The self-attention mechanism serves as a measure of similarity between the template and candidates in the search region. The integration of a template and search region feature extraction, along with the identification and elimination of background candidates, is accomplished through a single flow. Figure 3 illustrates the candidate elimination process. The self-attention mechanism in multiple vision transformer layers produces a feature map that is subsequently fed into a convolution neural network (CNN) in deeper layers. This enables the selection of the target position based on the highest classification score, facilitating the bounding box regression task.

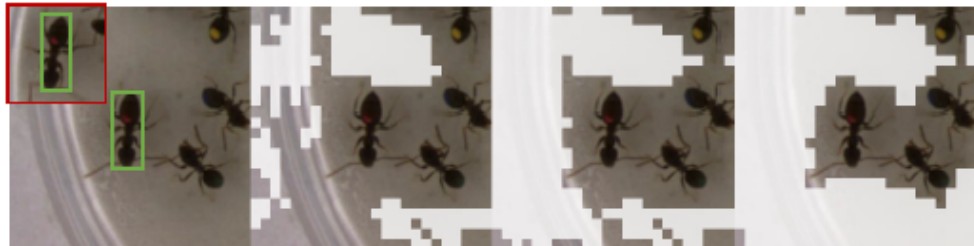

**Figure 3.** Example of background candidate elimination in OSTrack. The first image is the search region, the upper left corner image inside the first one is the template, and the green rectangles are the target objects.

The algorithm presented in that study was evaluated on various benchmark datasets, and it achieved state-of-the-art performance on the majority of them. One of the key advantages of the proposed method is its utilization of a single-stream network, which is computationally efficient and seamlessly integrated into real-time tracking systems.

One potential drawback is that the proposed algorithm relies on a substantial amount of training data to achieve accurate performance. Moreover, it may encounter challenges when tracking objects that exhibit rapid or irregular motion, such as those encountered in turbulent waters or high-speed vessel scenarios. Lastly, the computational demands

of the algorithm could pose limitations for real-time tracking applications, particularly in environments with limited resources.

This approach has the potential to be applied to maritime surveillance as it is designed for general object tracking tasks and does not have any specific limitations that would hinder its application in this domain. In maritime surveillance, the SOT approach could be a promising choice since the target object is typically a single vessel. The architecture of this work is described by the authors as "simple, pure, and effective" as it achieves state-of-the-art (SOTA) performance in all benchmark datasets analyzed, surpassing the current leading algorithms. Two significant advantages of this approach are the utilization of a transformer for visual processing and the incorporation of target background elimination. In the study by Ye et al. [48], it is demonstrated that the "Early Candidate Elimination" process, depicted in Figure 3, reduces computation, improves inference speed, and enhances overall performance in most scenarios.

### 4.2.2. SwinTrack: A Simple and Strong Baseline for Transformer Tracking

In their paper, Lin et al. [49] proposed a one-stage tracking algorithm called SwinTrack, which is based on transformers. The authors applied the Swin Transformer architecture, originally designed for computer vision tasks such as image classification, to the specific task of visual object tracking.

SwinTrack consists of three main steps: feature extraction using template and search region images, feature fusion, and prediction head. The authors of the proposed work employed different candidate network structures for various tracking components. The final stage focuses on predicting the bounding box target and performing intersection over union classification.

The initial step in SwinTrack involves utilizing a transformer backbone to facilitate weight sharing during the feature extraction process for both the template and search region. Each part generates its own feature tokens, which are subsequently concatenated to establish a connection for the subsequent step. This approach aims to minimize computation requirements by reducing the parameter count and by consolidating operations.

The second step of SwinTracks consists of two crucial components: the encoder and a decoder. The encoder calculates self-attention over concatenated tokens and feeds the result into a feed forward network to enhance the refinement of feature tokens. The decoder utilizes the output from the encoder as its input. By employing cross attention in the search region image, a feature map is generated. As suggested by the authors in [49], the combination of the encoder and decoder yields improved tracking performance and faster convergence.

The third and final step of SwinTrack involves a three-layer multi-layer perceptron. This MLP is responsible for performing classification and regression tasks, ultimately generating a bounding box in the target object in the tracking process.

The computational requirements and memory usage associated with training large-scale transformer models, like the one utilized in SwinTrack, could pose limitations for real-time tracking applications that have restricted computational resources. Furthermore, similar to other deep-learning-based approaches, the performance of SwinTrack might be influenced by the quality of input data, including occlusions, cluttered backgrounds, and variations in lighting conditions, which can potentially hinder its accuracy in object tracking.

SwinTrack has demonstrated its state-of-the-art performance on multiple benchmark datasets for visual object tracking, including OTB-2015 and LaSOT. The authors have further showcased the versatility of SwinTrack by fine-tuning the pre-trained model for other computer vision tasks such as object detection and segmentation. This adaptability positions SwinTrack as a robust foundation for future research on transformer-based object tracking, serving as a starting point for the development of more advanced tracking algorithms. Given its general-purpose nature, SwinTrack holds potential for application in various scenarios, including maritime surveillance.

### 4.2.3. MixFormer: End-to-End Tracking with Iterative Mixed Attention

In the work presented in [50], a novel approach called MixFormer was introduced, which offers advancements beyond the methodologies discussed in the previous two studies. Unlike traditional trackers that consist of multiple stages for feature extraction, integrating feature maps and target localization, MixFormer is an end-to-end visual object tracking algorithm that streamlines these process into a unified framework. Those three steps are well defined and present in almost all trackers.

MixFormer introduces a novel approach where feature extraction and feature mapping integration are combined into a single step. This integrated step effectively captures target information in a streamlined manner, allowing for the extraction of highly specific and discriminative features. By condensing the process into two steps, MixFormer achieves a lean architecture with a reduced number of parameters, resulting in improved computational efficiency.

The architecture of MixFormer incorporates a Mixed Attention Module (MAM), which combines feature extraction and information integration for the target template and search region. This module takes both images as input and utilizes self-attention mechanisms to facilitate target searches and to gather specific information. Following three stages of target searching, as described earlier, the resulting tokens are fed into a convolutional network responsible for determining the corners of the target bounding box.

It is worth noting that MixFormer performs exceptionally well on the LaSOT dataset, ranking third across all metrics. Furthermore, as a lean architecture, it offers significant parameter savings compared with related models. The authors conducted a comprehensive comparison, demonstrating that MixFormer achieves superior performance on the LaSOT dataset while requiring fewer parameters and less computational resources. The proposed approach was extensively evaluated on various challenging tracking benchmarks, including OTB-100, LaSOT, and TrackingNet, where it consistently outperformed state-of-the-art tracking algorithms in terms of accuracy and robustness.

MixFormer offers the advantage of being an end-to-end model, allowing direct training on tracking data without the need for pre-processing or post-processing steps. However, a potential drawback of the model is its computational complexity, primarily stemming from the iterative attention mechanism. This characteristic could impede real-time tracking performance on low-power devices that have limited computational resources.

This tracker is particularly valuable for maritime applications as it excels in general scenes. Being and end-to-end tracking algorithm, it has the potential to effectively track vessels and other maritime objects in real-time, while also being capable of handling common challenges encountered in maritime surveillance, such as motion blur and occlusion.

### 4.2.4. AiATrack: Attention in Attention for Transformer Visual Tracking

In their work [51], the authors introduce a novel architecture for object tracking that utilizes a transformer-based model. This architecture, referred to as AiATrack, extends the standard multi-head attention mechanism of the transformer by incorporating an additional "attention in attention" (AiA) module. Similar to many contemporary trackers, the feature extraction step of AiATrack incorporates transformer attention. The attention process involves computing the weighted sum of key–value pairs to generate output. The query, key, and value vectors are derived from input vectors and used as input to the attention module. The goal is to map the query and the pairs to produce an output. The resulting output is weight-dependent, with the weights determined by the correlations between queries and keys using a compatibility function. Further details on the attention mechanism commonly employed in this context can be found in [52].

The design of AiATrack was motivated by an analysis of the correlations between query–key pairs. This analysis revealed that the correlations, when considered in isolation, may lead to erroneous associations due to various factors in the image field, such as background distractions. To address this limitation, the AiA module is introduced in the architecture.

The AiA module aims to enhance correlation consistency among query–key pairs, thereby improving the correlations for new query–key pairs and suppressing irrelevant ones. Another notable improvement in the AiA tracker is the transformer decoder. Unlike many trackers that rely on a fixed reference image for the search region, the AiA tracker updates the reference image during tracking to account for potential target location drift. This is achieved by performing cross-attention not only between the reference (initial) frame and the current frame but also between the reference frame and a neighboring frame in the sequence.

The final step of this tracker involves predicting the corners of the bounding box. A convolutional network is employed to generate probability maps for each corners, including the left, right, up, and down corners. The accuracy of the corner prediction is evaluated using the intersection over union metric with the ground truth bounding box. To incorporate all the features within the bounding box, a polling layer is utilized, and the resulting output is then fed to a fully connected network for further processing.

The proposed architecture demonstrates superior performance compared with other state-of-the-art trackers across various tracking benchmarks, including LaSOT [53]. It achieves an AUC score of 0.607, surpassing the results obtained by other leading trackers in the field.

AiATrack has a few noteworthy limitations. Firstly, its computational requirements are relatively high compared with certain tracking architectures, potentially restricting its real-time applicability in certain scenarios. Secondly, the algorithm's performance heavily relies on the availability of a substantial amount of training data, which may pose challenges in domains where acquiring such data is difficult. Additionally, similar to other transformer-based tracking approaches, AiATrack may encounter difficulties in handling occlusions and abrupt changes in object appearance or motion. Lastly, while the attention-in-attention mechanism employed in AiATrack facilitates capturing spatial and temporal dependencies, it can be sensitive to noisy input data and may necessitate careful fine-tuning for optimal performance.

The authors suggested that AiATrack has the potential to be applied not only in object detection in videos but also in MOT tasks. Furthermore, AiATrack shows promise in the field of maritime surveillance. Given its visual tracking capabilities, the algorithm could be utilized to track ships or other objects of interest in video sequences captured by cameras mounted on maritime vessels or stationed on land-based observation points.

### 4.2.5. Unicorn—Towards Grand Unification of Object Tracking

In [4], the authors presented a network architecture designed to address the four primary tasks in tracking: SOT, MOT, Video Object Segmentation (VOS), and Multi-Object Tracking and Segmentation (MOTS). They proposed a unified framework that can accommodate various tracking methods, including deep-learning-based approaches, correlation filters, and graphical models. The framework consists of three main components: a visual model, a motion model, and a data association model. The visual model captures the target object, while the motion model predicts its trajectory over time. The data association model is responsible for assigning detections to existing tracks or initiating new tracks when a target is detected.

The authors named the framework Unicorn, and it was designed to learn various data sources in tracking and to address all four tasks using the same model parameters. Unlike previous approaches, Unicorn considers both the reference frames and the current frames as complete images, rather than focusing solely on the region of interest. This choice allows for target re-detection in the case of disappearance. The network backbone is fed with the entire images, providing target information and generating a feature map. The approach varies for different tasks: for SOT and VOS, pixel-level correspondence is established across the entire input image, while for MOT and MOTS, the tracker involves $M$ trajectories in the reference frame and $N$ instances in the current frame. In this case, the correspondence is made at the instance level rather than the pixel level.

It is worth noting that this work introduces a novel approach utilizing the deformable attention module instead of the traditional transformer attention mechanism. This deformable attention module offers improved efficiency in terms of memory consumption, particularly when dealing with long sequence lengths. For the MOT task, the objective is to detect objects of various classes and to match them to their respective category. In contrast, the SOT task aims to detect any target within the reference frame. The final process employs two different approaches: Target Priors, which are utilized for SOT and VOS, and Feature Fusion, which is employed in all scenarios. It is worth mentioning that SOT and VOS with non-zero prior differences can lead to more focused tracking of the targets.

The authors validated the efficacy of their framework through extensive experiments on various tracking benchmarks, showcasing its superiority over state-of-the-art trackers. Furthermore, they offered an open source implementation of their framework, which serves as a valuable foundation for future research in the field of object tracking.

One notable drawback is the substantial requirement for a large volume of training data, which may not always be accessible or viable to gather. Another limitation is the high computational cost associated with the framework, rendering it less practical for real-time applications. Moreover, the performance of the framework may be susceptible to variations in the object's appearance or behavior, potentially resulting in tracking failures or inaccuracies. Finally, the proposed framework does not address the issue of effectively handling occlusions, which poses a significant challenge in the realm of object tracking.

The framework holds potential for application in maritime surveillance, particularly in scenarios involving simultaneous tracking or detection of multiple ships within the field of vision. By unifying diverse tracking methods and datasets, the framework establishes a cohesive tracking paradigm that can prove advantageous for monitoring objects in maritime surveillance settings. The inclusion of techniques like online learning, model adaptation, and long-term tracking, the framework enhances its ability to track ships and other objects effectively in maritime environments.

### 4.2.6. STMTrack: Template-Free Visual Tracking with Space-Time Memory Networks

STMTrack, introduced in [54], draws inspiration from human visual memory to sustain the identity of tracked objects in a continuous manner over time. This deep-learning-based algorithm for visual object tracking incorporates a space–time memory (STM) module to capture the extended temporal dependency of object appearances. By employing the STM module, the tracker can retain a memory bank of target object representations throughout the tracking process, enabling more precise tracking. Moreover, this tracker demonstrates effectiveness even in the presence of consistent environmental variations.

The algorithm operates in a template-free manner, implying that it does not rely on a pre-defined target template for tracking purposes. Instead, it utilizes the object's bounding box in the initial frame as a starting point and learns the target's appearance and motion patterns in an online fashion. STMTrack comprises a backbone network for feature extraction, a space–time memory network for storing and retrieving feature extraction, and a target model for updating the object's appearance and motion patterns. Additionally, it incorporates a dynamic feature aggregation mechanism to fuse features from various layers of the backbone network, enhancing the tracking performance.

The first step of feature extraction involves considering $T$ frames as memory and $T$-label maps in the first plane. The inclusion of the first plane map is crucial to mitigate background distraction during target feature learning. Since we have $T$ frames in "memory", we consequently have $T$ features maps in "memory" as well. The second step entails computing similarities between each pixel in both feature maps, resulting in a similarity matrix. This step enables the recovery of the target information by leveraging multiple frames from memory. In STMTrack, the feature extraction stage does not separate the extracted features into keys and values but directly utilizes them for target localization. This approach enhances the suitability of space–time net for SOT tasks. The final step involves classification and regression to discern the target from the background and to estimate the

target's bounding box. In the feature extraction and head network, which encompasses the first and third components of the proposed method, the authors employed convolutional layers to reduce the dimensionality of features and the output from the classification and regression task. In convolutional layers, kernels are defined, and the output is obtained through the convolution operation between the kernel and input array. It should be noted that the input array can be an image or any other component within the architecture.

The proposed method includes a significant component known as the space–time memory net, which enables the storage of multiple frames of target information within a memory structure. Each frame contributes to the overall memory, granting the tracker robust adaptability to variations in the target. This adaptive capability is highly desirable in practical scenarios where controlled environments for frame capture are often unattainable. The algorithm is thoroughly evaluated on various challenging tracking benchmarks, such as OTB-2015 and LaSOT, and consistently exhibits superior performance compared with other state-of-the-art trackers.

One notable drawback of STMTrack is its high memory requirement for storing feature maps of previous frames, potentially hindering its performance on low-memory devices. Moreover, being a template-free tracker, it may face challenges with objects or scenes lacking distinctive visual information to differentiate the target from the background. Lastly, its accuracy on fast-moving targets may be comparatively lower than that of some other state-of-the-art trackers, as it heavily relies on spatial and temporal memory.

STMTrack holds potential for application in maritime surveillance due to its versatility as a general-purpose visual tracking algorithm, capable of tracking any detectable and localized object in video frames. Given that ships are typically detected or tracked once, the space–time nature of STMTrack makes it a particularly intriguing choice for such tasks in maritime surveillance.

### 4.2.7. KeepTrack and KeepTrackFast-Learning Target Candidate Association to Keep Track of What Not to Track

In [55], the authors introduced the KeepTrack algorithm to tackle two common challenges in object tracking: the presence of distractors that closely resemble the target object and significant variations in the target object's appearance over time. KeepTrack addresses these challenges through the use of a candidate proposal mechanism, which generates a set of potential object candidates, and a candidate association module, which assigns each candidate to either the most likely target object or a distractor. By sequentially learning the association between the candidates and the target object, KeepTrack can effectively distinguish distractors and adapts to changes in target appearance. The primary objective is to maintain a considerable distance between the tracker and distractors. Additionally, the authors introduced a faster variant of KeepTrack, known as KeepTrackFast, which incorporates subtle changes to improve runtime performance. For instance, the number of bounding box refinement steps was reduced from 10 to 3. Moreover, the target candidate extraction step employs a different threshold that disregards candidates with low target classifier scores, resulting in the processing of fewer frames with multiple candidates. The subsequent paragraphs exclusively discuss the details of KeepTrack, while Table 3 presents the relevant information separately.

The architecture of the KeepTrack algorithm involves several steps. Firstly, a target score map is predicted for the current frame, and the target candidate with the highest score is extracted. It is worth noting that the association is established between the current frame and the preceding one. The next step involves the extraction of target candidates along with their respective coordinates. For each constructed candidate, a feature set is generated based on similarities in location and distance to different objects, while also considering slight variations in appearance among objects. The authors clarify that these extracted features play a crucial role in establishing mappings related to appearance, target similarity, and position.

The scoring mechanism plays a pivotal role in this approach. In case of occlusion, the object may become erroneously associated with a distractor. However, if the object's score consistently increases over time, it is more likely to be selected as the target in the future frames. The primary objective is to track both the target object and potential distractors. KeepTrack incorporates a mechanism to effectively handle distractor objects by training a separate classifier that can differentiate between target candidate between target candidate and distractor candidates.

The algorithm undergoes evaluation on various demanding benchmarks for object tracking, such as OTB-2015 and LaSOT. The results showcase its superior performance compared with other state-of-the-art trackers.

The KeepTrack algorithm offers several advantages, including its capability to effectively manage visually similar objects as distractors and to handle significant changes in the target object's appearance over time. Furthermore, it has the ability to track multiple objects simultaneously, which is crucial for various real-world scenarios. Leveraging deep learning techniques enhances its accuracy and robustness, surpassing traditional tracking methods.

However, the KeepTrack algorithm does have certain limitations. Firstly, it assumes that the object of interest remains visible throughout the entire sequence, which may not always hold true in real-world scenarios. Secondly, the algorithm heavily relies on the quality of object proposals and their association with the target object, which can be a challenging task in cluttered and complex scenes. Thirdly, the algorithm's performance is highly dependent on the availability of a large amount of training data, which may pose a limitation in certain applications. Lastly, the algorithm does not explicitly model long-term temporal dependencies, which can result in tracking drift over extended sequences.

This work was included in this review due to its strong performance across all analyzed datasets. Additionally, the algorithm's capability to handle visually similar objects and cope with significant appearance changes over time could be advantageous for tracking ships or other vessels, which often exhibit similar appearances or undergo variations in appearance due to factors such as weather conditions. However, the effectiveness of the algorithm would ultimately rely on the quality of the input data and the specific characteristics of the maritime surveillance application at hand.

### 4.2.8. AAA: Adaptive Aggregation of Arbitrary Online Trackers with Theoretical Performance Guarantee

In the paper by Song et al. [56], a method is introduced to enhance the accuracy of visual object tracking by combining multiple online trackers in real time. The proposed approach, known as adaptive expert aggregation (AEA), focuses on aggregating predictions from these trackers to generate a consolidated solution. In AEA, each expert corresponds to an online tracker that provides its own estimation of the target location as an individual solution. Consequently, for every frame $t$, $N$ diverse online trackers produces $N$ location predictions, which are then merged into a single solution for that specific frame.

The authors employ the EAE performance theory warranty to establish the strength of AAA. However, ensuring accurate weight updates for the experts in online tracking scenarios poses a challenge. To tackle this issue, the proposed method introduces the use of an anchor frame and subsequent feedback to update the expert weights. If an anchor frame is unavailable, the weights remain unchanged. An anchor frame is defined as a frame wherein the target location exhibits a high confidence value.

The proposed method functions by assigning weights to each tracker, which are determined by their historical performance, and then combines the tracker outputs using a weighted average. The weights of the trackers are dynamically updated based on their current performance and their significance in contributing to the overall tracking accuracy. The method aggregates $N$ different trackers, and the weights are updated using the anchor frame and the subsequent feedback. The performance guarantee of AAA is defined utilizing the EAE theory.

The proposed method enhances tracking accuracy and robustness by effectively combining multiple online trackers, accommodating any arbitrary number of trackers. The method offers a theoretical performance guarantee based on the EAE theory and operates in real-time, making it well-suited for tracking applications that demand prompt and accurate outcomes. Moreover, the algorithm demonstrates flexibility and reliability by utilizing an anchor frame and subsequent feedback to update the weights of expert trackers when there is a satisfactory level of confidence in the target's location.

The model experiments presented in the paper showcase that the proposed method attains state-of-the-art performances in terms of accuracy and robustness, even in scenarios where some of the expert trackers encounter failures. However, it is worth noting that the model utilizes a combination of 12 expert trackers, which can incur significant computational costs. Additionally, the training and evaluation process for multiple expert trackers can be time-consuming and resource-intensive.

In certain application scenarios, the cost associated with employing multiple expert trackers may not be justifiable. Nevertheless, the proposed method can still prove valuable in such cases by leveraging crucial information from the application domain to establish the anchor frame and to ensure a high level of reliability.

On the other hand, the method could find application in maritime surveillance. Despite the potentially high computational cost associated with combining multiple online trackers, the utilization of anchor frames and later feedback can contribute to the reliability of the tracking output. Moreover, in the context of maritime surveillance, the cost of combining multiple trackers may be justified, and the theoretical performance guarantee provided by the EAE theory could serve as a means to evaluate the quality of the tracking results.

### 4.2.9. ASRCF Model-Visual Tracking via Adaptive Spatially Regularized Correlation Filters

The authors of [57] proposed an efficient and robust method for object tracking that utilizes correlation filters to estimate the location of the target. Additionally, they incorporate adaptive spatial regularization to enhance the discriminative power of the filter.

The proposed method operates in two stages. In the first stage, the target's location is initialized in the first frame, and an Adaptive Spatially Regularized Correlation Filter (ASRCF) is trained to estimate the target's location in subsequent frames. The ASRCF is trained using an L2-regularized regression framework, which learns linear mapping between the filter and the target's appearance. This trained filter is then applied to the subsequent frames to predict the target's location.

In the second stage, the method incorporates adaptive spatial regularization to enhance the discriminative capability of their filter. This is motivated by the idea that the filter's response should exhibit spatial smoothness across the entire image. To enforce this constraint, the method introduces an adaptive spatial regularizer that adapts based on the local characteristics of the image. For each pixel in the image, a regularization term is computed based on its proximity to the predicted location of the target. This regularization term is subsequently employed to modify the response of the filter, thereby yielding a more refined and resilient response.

The proposed method offers several advantages. Firstly, it utilizes the ASRCF, which enables the filters to learn adaptable shapes and to enhance their capability to handle target object variations, including deformation, rotation, and scaling changes. This adaptability proves particularly valuable in maritime scenarios. Secondly, the model incorporates a filter penalty mechanism that improves reliability when dealing with occlusions. Additionally, the method leverages the Fourier transform and employs multiple levels of feature combination, which contribute to precise estimation of the object's location.

However, there are also certain drawbacks to this approach. One limitation is its strong dependence on the accuracy of the initial bounding box, which can lead to poor performance when the initial bounding box is imprecise. Furthermore, the model's effectiveness may be compromised by variations in the target's appearance caused by factor such as changes in illumination or occlusions. Lastly, the computation complexity of the approach can be

significant, particularly in the context of large-scale maritime scenarios, thereby restricting its practical applicability.

The proposed method underwent evaluation on various challenging datasets, show-casing its achievement of state-of-the-art tracking performance in accuracy and robustness. Notably, the method is computationally efficient, rendering it suitable for real-time applications. Moreover, the adoption of adaptive spatial regularization enables the approach to effectively handle deformations, occlusions, and other challenging scenarios, thereby solidifying its potential as a promising approach for visual tracking.

### 4.2.10. SiamMask-Fast Visual Object Tracking with Rotated Bounding Boxes

The paper [3] introduces a visual tracking algorithm that focuses on tracking rotated bounding boxes. The authors present a novel approach to address the challenge of real-time tracking of rotated objects. The algorithm utilizes correlation filtering and is specifically tailored to track objects with arbitrary rotations.

The algorithm consists of several stages. Firstly, it initializes the tracker by using the first frame of the video sequence. It extracts the features of the object and generates a rotated bounding box around it. The features are represented using color histograms and local binary patterns. Subsequently, the algorithm constructs a filter model based on these features, which is utilized to track the object in the following frames.

The tracking process operates by convolving the filter model with the features of the current frame, resulting in a response map that indicates the probable location of the object in the current frame. Subsequently, the algorithm utilizes the response map to update the state of the tracker by computing a new rotated bounding box around the object.

The proposed method offers several advantages compared with previous tracking algorithms. The utilization of rotated bounding boxes enables accurate tracking of objects with arbitrary orientations in real time, rendering it applicable to various scenarios. Moreover, the incorporation of correlation filters enhances the algorithm's resilience to variations in lighting and appearance. Furthermore, the algorithm exhibits computational speed, efficiency, and effectiveness, making it well suited for real-time applications.

However, the proposed method does have certain limitations. Firstly, it is heavily reliant on the accuracy of the object's initial bounding box, and therefore, its performance may suffer if the initialization is imprecise. Secondly, the method may struggle in scenarios where the object undergoes significant appearance changes or experiences occlusion by other objects in the scene. Lastly, the evolution of the method's performance primarily focuses on object tracking datasets, while its effectiveness in other tasks such as object detection or instance segmentation remains unexplored.

The algorithm was evaluated on multiple benchmark datasets and was compared with state-of-the-art tracking algorithms. The results indicate that the proposed method surpasses previous approaches in terms of both tracking accuracy and speed. Furthermore, the authors showcased the algorithm's efficacy in tracking objects with intricate rotations in real-world scenarios, including the tracking of cars in traffic.

In maritime scenarios, ships often exhibits various orientations, making the utilization of rotated bounding boxes more suitable for accurately representing their shape and movement compared with axis-aligned bounding boxes. Moreover, the proposed method specifically caters to real-time tracking requirements, which proves beneficial for monitoring moving ships in real-world settings. However, it is important to note that the algorithm's performance could be influenced by environmental factors including lighting conditions, weather variations, and sea conditions.

### 4.2.11. SiamVGG: Visual Tracking Using Deeper Siamese Networks

The authors of [58] propose a SOT method based on a deep neural network called the Siamese network. This method utilizes two networks, namely the feature extractor network and the similarity network, to track an objective in a video sequence. The feature extractor network is implemented using a VGG-16 architecture, which is pre-trained on the

ImageNet dataset. The similarity network takes the feature maps from the two branches of the feature extractor network and generates a scalar similarity score, indicating the level of similarity between the target object and the search region in the current frame.

During the training process, a pair of images is randomly selected from the dataset, and the network is trained to minimize the disparity between the similarity score and a binary label that denotes whether the two images depict the same object. This training process is conducted offline, eliminating the need for online updates contributing to the computational efficiency of the method.

In the testing phase, the target object is initially localized in the first frame, and its appearance is utilized to create a template. In each successive frame, the search region encompassing the anticipated position of the target object is inputted into the Siamese network, and the similarity score is computed. The estimated location of the target object in the current frame is subsequently determined based on the highest similarity score.

The authors additionally introduce a spatially weighted average pooling method to enhance the accuracy of their approach. This method calculates the weighted average of the feature maps using the similarity scores as weights. The weights are determined by a Gaussian kernel, which assigns greater weights to the feature maps in closer proximity to the predicted location of the target object.

The algorithm presents some advantages. First, the use of a deeper architecture can provide higher accuracy compared with previous methods. Second, the use of a Siamese network allows the model to learn from the limited data available, improving generalization and robustness to variations in target appearance. Lastly, the model is efficient and can run in real time on standard hardware, which is useful for practical applications.

However, there are several limitations to this method. Firstly, it may struggle to handle significant changes in appearance or pose, as it does not incorporate 3D structure or temporal consistency information. Secondly, obtaining a substantial amount of training data to effectively learn the features can be challenging, particularly for certain target objects or environments. Additionally, the use of fully convolutional Siamese networks can render the model sensitive to minor shifts in the target object, potentially resulting in drift during tracking. Lastly, the method may not handle occlusions or background clutter adequately, leading to tracking failures in such scenarios.

The experiments presented in the paper provide evidence that the proposed method outperforms several state-of-the-art tracking methods in terms of both accuracy and speed, as demonstrated on four standard tracking benchmarks. Furthermore, the method exhibits resilience to occlusions and deformations, thereby rendering it appropriate for real-world applications.

Being a visual tracking algorithm, it can be employed for real-time ship tracking using video sequences captured by cameras installed on ships or coastal areas. Furthermore, owing to its foundation on a deep convolutional neural network, the algorithm possesses the capability to learn and adjust to the target object's appearance, even amidst variations such as changes in illumination, occlusion, or other factors.

Similar to other visual tracking algorithms, the performance of SiamVGG may be influenced by environmental factors like heavy sea conditions or fog, resulting in blurry or noisy images. Additionally, the size of the target object, its speed, and trajectory can impact the algorithm's accuracy, thus affecting the tracking quality.

### 4.2.12. SE-SiamFC-Scale Equivariance Improves Siamese Tracking

The study by Sosnovik et al. [59] presented a novel approach to enhance the performance of Siamese-based visual object tracking. The proposed method extends existing Siamese tracking algorithms by introducing a scale-equivariant Siamese network, which enables the network to automatically adapt to changes in the target's scale. In this context, the SE-SiamFC was introduced as a variant of SiamFC that incorporates scale equivariance. The Siamese architecture is a widely used technique for visual object tracking, relying on a comparison of the target object features with features extracted from candidate patches in

subsequent frames. However, a key challenge faced by Siamese-based tracking method is their limited ability to track objects at varying scales, which can hinder their performance in real-world scenarios. To address this limitation, the proposed network consists of two identical subnetworks that share weights and is trained using a multi-scale sampling strategy, enhancing the network's scale robustness.

To tackle this challenge, the authors presented a novel approach that incorporates scale-equivariant features into the Siamese network. These features were specifically designed to ensure consistent responses across various scales, enabling the network to effectively track objects at different scales with enhanced accuracy. This is accomplished through architectural modifications to the Siamese network, including the integration of a scale-equivariant module that adapts the network's feature to match the scale of the tracker object.

This work offers several notable advantages. Firstly, it achieves scale invariance without requiring explicit scale estimation or additional modules, resulting in a streamlined and efficient tracking system. Secondly, the proposed method demonstrates robustness in challenging scenarios, such as variations in illumination, occlusion, and motion blur. Furthermore, the proposed method exhibits computational efficiency, rendering it well suited for real-time applications.

However, the proposed method may face limitations in scenarios where the target experiences substantial scale changes that fall outside the training range of the network. Additionally, training the network necessitates acquiring extensive annotated datasets, which can be challenging in certain scenarios. Lastly, the method's performance may be less optimal for object classes exhibiting complex deformations, thus reducing its suitability for certain applications.

The proposed method undergoes evaluation on various benchmark datasets, where it exhibits superior performance compared with existing state-of-the-art Siamese-based tracking methods in terms of accuracy and speed, particularly when dealing with objects that undergo significant scale changes during tracking. Furthermore, the authors provide evidence of the method's effectiveness in real-world scenarios, including tracking pedestrians in crowded urban environment and tracking ships in satellite imagery.

The capability to track objects despite scale variations is of great importance in maritime surveillance scenarios, where objects in the sea can exhibit varying distances and sizes. The proposed method addresses this challenge by enhancing Siamese trackers through the incorporation of scale-equivariant convolutions and the integration of scale adaptation into the feature extraction process. These improvements render the tracker more robust to scale changes and enhance its capacity to track objects at diverse scales. As a result, the method holds promise for maritime surveillance applications, where scale variations frequently occur.

### 4.2.13. BoT-SORT: Robust Association Multi-Pedestrian Tracking

In [60], the authors introduced a multi-object tracking algorithm called BoT-SORT, which is specifically designed for multi-pedestrian tracking. BoT-SORT builds upon the well-known object tracking framework known as Simple Online and Realtime Tracking (SORT). It incorporates two key enhancements: a batch update strategy and an orthogonal transformation to handle complex scenarios. The algorithm employs a bipartite graph optimization technique for data association and incorporates a Kalman filter for trajectory prediction and smoothing.

The algorithm begins by employing a deep neural network to detect and track pedestrians in each frame. Subsequently, a batch processing strategy optimizes the associations between the detected pedestrians across multiple frames, thereby enabling BoT-SORT to effectively handle occlusions and to enhance tracking accuracy.

To achieve this, the algorithm initially performs a batch update by associating the detected pedestrians across multiple frames using a similarity matrix. This matrix is calculated based on various features, such as the location and appearance of the bonding

boxes. BoT-SORT then applies an orthogonal transformation to refine the associations, thereby mitigating the impact of occlusions and other tracking challenges. In addition, BoT-SORT incorporates several other features to enhance tracking accuracy and robustness. For instance, it incorporates a track management system that handles tasks such as track initialization, termination, and fragmentation. Moreover, the algorithm includes a mechanism to address missed detections and to re-identify pedestrians who temporarily exit the scene.

BoT-SORT offers several advantages. Firstly, it demonstrates the ability to handle complex real-world scenarios characterized by high pedestrian density, occlusion, and re-identification challenges. Secondly, it surpasses the state-of-the-art performance on multiple datasets, achieving high scores in terms of MOTA and IDF1. Lastly, it exhibits computational efficiency, enabling real-time tracking even on a single CPU.

However, there are also some limitations to consider. Firstly, BoT-SORT is specifically designed for tracking pedestrians and does not extend its capability to track other objects or vehicles. Secondly, it relies on pre-detection of pedestrians, which may restrict its applicability in certain scenarios where detection is challenging or unavailable. Lastly, it may face difficulties in crowded environments with substantial occlusion and limited distinguishable features among pedestrians.

The algorithm was thoroughly evaluated on various public datasets, and the results demonstrated its superior performance compared with other existing state-of-the-art multi-object tracking algorithms. It excelled in terms of tracking accuracy and computational efficiency. The effective combination of Bayesian optimization, bottom-up data association, and state-space modeling makes it highly suitable for real-world applications, including video surveillance and crowd monitoring.

The application of the BoT-SORT algorithm to maritime surveillance holds potential, but it would necessitate tailored modifications and adaptations to account for the specific characteristics and challenges of this domain. For instance, the algorithm could be customized to detect and track ships instead of pedestrians, accommodating the distinct movement patterns and sizes of vessels compared with humans. Additionally, the algorithm may require adjustments to effectively handle occlusions resulting from obstacles like other ships, waves, or weather conditions. In essence, the BoT-SORT algorithm offers a resilient framework for multi-object tracking that can be flexibly adapted to diverse surveillance applications, including those encountered in maritime scenarios.

### 4.2.14. StrongSORT: Make DeepSORT Great Again

The paper [61] introduced a novel multi-object tracking algorithm that enhances the performance of the popular DeepSORT (deep-learning-based SORT) algorithm. The primary objective of this research is to overcome the limitations inherent in DeepSORT and to deliver a more robust and accurate solution for object tracking.

The main contribution of the StrongSORT algorithm lies in its adoption of a new tracking framework that incorporates a powerful CNN feature extractor known as the Strong CNN. This deep neural network is pre-trained on a vast dataset of images, enabling it to extract high-level features that are highly relevant for object tracking. By utilizing the Strong CNN as a feature extractor, the algorithm generates embeddings for each detected object, facilitating object matching across frames and enabling robust tracking over extended periods of time.

Alongside the Strong CNN feature extractor, StrongSORT incorporates various other enhancements compared with DeepSORT. One notable improvement is the introduction of an adaptive tracklet length that can be dynamically adjusted according to the motion patterns exhibited by objects in the scene. This adaptive approach enables the algorithm to effectively handle objects with different speeds and accelerations, enhancing its tracking capabilities in diverse scenarios.

Another enhancement introduced in StrongSORT is the utilization of a multi-scale approach to object detection and feature extraction. This approach involves detecting

objects at various scales and employing multiple feature extractors to generate embeddings for each scale. By incorporating this multi-scale strategy, StrongSORT enhances the accuracy of object detection and tracking, particularly for small or distant objects that may require more precise analysis.

Finally, the authors of StrongSORT also introduce a novel gating mechanism that improves the reliability of object associations. This gating mechanism uses a learned function to score the likelihood of each potential object association and only allows associations with high scores to be used in the tracking process. This helps to reduce the number of false positive associations and to improve the overall tracking accuracy.

Lastly, the authors of StrongSORT introduce a novel gating mechanism aimed at enhancing the reliability of object associations. This gating mechanism utilizes a learned function to assign scores to potential object associations, allowing only associations with high scores to be utilized in the tracking process. By incorporating this gating mechanism, the algorithm effectively reduces the occurrence of false-positive associations, leading to improved tracking accuracy overall.

One of the primary contributions of StrongSORT is the introduction of a novel online learned affinity metric. This metric enables the tracker to update its matching criteria dynamically and to adapt to the appearance changes of tracked objects. This adaptive approach enhances the algorithm's robustness in handling variations in object appearance. Moreover, StrongSORT integrates re-identification with tracklet generation, resulting in a more efficient and robust process, particularly when dealing with occlusions. Additionally, the algorithm incorporates a feature selection mechanism that effectively reduces the dimensionality of the feature space, leading to improved tracking accuracy.

However, there are also some disadvantages to consider. One of the main drawbacks of the StrongSORT algorithm is the high computational resources required for training the deep neural network and making predictions. Another potential limitation is that the algorithm relies on appearance features, which can be susceptible to issues such as illumination changes, occlusions, and other visual distortions. Lastly, although StrongSORT has demonstrated improved accuracy compared with DeepSORT, it may still encounter challenges when dealing with crowded scenes or scenes with a high density of objects.

The algorithm represents a substantial improvement over DeepSORT and offers a highly accurate and robust solution for object tracking in complex scenes. It attains state-of-the-art performance on various tracking benchmarks, surpassing not only DeepSORT but also other recent tracking methods. Its advanced feature extraction capabilities and adaptive tracking framework render it well suited for a broad range of applications encompassing surveillance, robotics, and autonomous driving.

The algorithm has the potential to be applied to maritime surveillance since it is a versatile multi-object tracking algorithm capable of handling different types of objects, including those found on the water. However, the feasibility of applying the algorithm to maritime surveillance tasks would depend on specific requirements and challenges, such as the size and movement characteristics of the tracked objects, lighting and weather conditions, and the presence of occlusions or other visual distortions. Additionally, when evaluating its suitability for the task, considerations should be given to the algorithm's accuracy and computational requirements.

### 4.2.15. SimpleTrack: Rethinking and Improving the JDE Approach for Multi-Object Tracking

The paper introduces SimpleTrack, a multi-object tracking algorithm presented in [62]. SimpleTrack aims to enhance the performance of the widely used Joint Detection and Embedding (JDE) approach by implementing several modifications. The core concept of SimpleTrack is to streamline the JDE approach, resulting in improved efficiency and accuracy.

The SimpleTrack algorithm introduces a decoupled approach for object detection and appearance identification, which differs from the JDE approaches that utilize a single network for both tasks. SimpleTrack utilizes the DLA-34 architecture as the backbone for

extracting feature maps and incorporates two distinct fusion methods: up-to-bottom fuse for detection and bottom-to-up fuse for re-identity. The authors highlight the challenges of using intersection over union (IoU) alone in occlusion scenarios or when the target loses its ID, which can make ID retrieval difficult. To address this, the algorithm introduces JDE-based methods that mitigate the issue. Additionally, when the ID loss duration becomes significant, accuracy may decline when using Kalman-filter-based predictions.

To address these challenges, the authors proposed a solution called SimpleTrack, which incorporates similarity matrices, specifically the EG Matrix-Embedding and Giou Matrix. The EG Matrix utilizes the cosine distance to track long-range targets, while the Giou Matrix helps to limit correspondence gaps. These matrices are constructed using information such as object location, movement, and appearance. The EG matrix combines the location matrix, which represents the Giou matrix based on bounding box information, and the appearance distance matrix, which measures the cosine distance between different appearance embedding vectors. The final EG matrix is obtained by linearly combining these two matrices. Notably, the EG matrix itself, even without the SimpleTrack algorithm, is a valuable contribution to the development of new models, as demonstrated through its evaluation with other JDE-based approaches.

The algorithm has several advantages. Firstly, it has demonstrated improved performance compared with the original JDE algorithm across several benchmark datasets. SimpleTrack achieved a better balance between accuracy and efficiency, particularly in crowded scenes with a high object density. This improvement was achieved by enhancing the quality of detection and re-identification strategies, as well as by introducing a more efficient similarity metric. Secondly, SimpleTrack employed a simple yet effective approach to handle occlusions and crowded scenes. Instead of relying on complex models or techniques, the algorithm utilized straightforward heuristics and assumptions. For example, it leveraged the observation that occluded objects tend to move slowly and that objects in close proximity are more likely to belong to the same category. This simplicity not only enhanced interpretability but also facilitated easier implementation. Finally, SimpleTrack achieved its improved performance with fewer computational resources compared with the original JDE algorithm. This was accomplished through code optimization, a reduction in model parameters, and the introduction of an efficient GPU implementation. These optimizations made the algorithm more practical and accessible for real-world applications.

Despite its advantages, SimpleTrack also has some limitations. One of the main drawbacks is its dependence on the object detection module, which can be susceptible to the quality and accuracy of the input images. This reliance introduces the possibility of tracking failures or inaccurate predictions, particularly in challenging or noisy environments. Moreover, the algorithm still relies on appearance features, which are vulnerable to illumination changes, occlusions, and other visual distortions.

Overall, SimpleTrack enhances the tracking performance of the JDE approach by simplifying the detection network and introducing a new tracklet-level feature aggregation approach. The experimental results demonstrate that SimpleTrack surpasses the JDE approach across multiple benchmark datasets.

The algorithm is well-suited for maritime surveillance applications. It is designed to detect and track multiple objects in real time, making it particularly applicable in environments with a high density of vessels or other objects in a marine setting. Furthermore, its capability to handle occlusions and track objects over long distances makes it a valuable tool for effective maritime surveillance.

However, it is important to consider that the algorithm's performance may be influenced by specific factors present in a maritime environment, such as the presence of waves or varying weather conditions. These factors can affect the appearance and movement of the tracked objects, potentially leading to tracking challenges. Potentially leading to tracking challenges, the performance of the entire system would be heavily influenced by the quality of the detection module, which may pose some difficulties in the marine environment.

### 4.2.16. GTR (Global Tracking Transformers)

Global Tracking Transformers (GTR), proposed by Zhou et al. in 2021 [63], is a state-of-the-art multi-object tracking algorithm. It introduces a novel tracking methodology that harnesses the power of transformers, a deep neural network architecture known for its exceptional performance in natural language processing and computer vision tasks. The core concept behind GTR involves conducting multi-object tracking through a global attention mechanism that operates on entire frames rather than processing individual objects separately. This distinctive approach sets it apart from conventional methods that rely on a detection and association pipeline for tracking objects individually. By addressing the challenge of associating objects across extended temporal gaps, where appearance changes and occlusions may occur, GTR aims to enhance tracking accuracy and robustness.

GTR employs an encoder–decoder architecture based on transformers to capture the interdependencies among all object pairs within a scene. This global modeling approach enables the algorithm to consider all objects simultaneously, leading to enhanced tracking performance, particularly in challenging scenarios characterized by numerous occlusions and object interactions. Given a sequence of RGB frames as input, GTR generates predicted object trajectories as its output. The encoder module processes each frame to extract spatial and temporal information, producing a set of features. Subsequently, the decoder module utilizes these features to generate object detections and their corresponding trajectories.

GTR possesses a notable advantage in its capability to conduct online tracking without the need for pre-processing tasks like object detection or feature extraction. This is accomplished by leveraging a self-attention mechanism, which enables GTR to learn to focus on pertinent regions within the input frames while disregarding irrelevant or noisy data. Moreover, GTR incorporates a memory module that permits the retention of information regarding previously observed objects, enabling effective tracking across extended temporal gaps.

GTR offers an additional advantage in its capability to manage occlusions and to track objects across multiple camera views. This is achieved through the acquisition of a view-invariant object representation that can be shared across various camera perspectives. Consequently, GTR can effectively track objects across different camera views and address occlusions by predicting the movement of occluded objects using their prior trajectories.

While GTR offers notable advantages, it also has certain limitations to consider. Firstly, the algorithm's high computational cost can hinder its suitability for real-time applications on low-power devices. Secondly, GTR's dependency on a substantial volume of training data may pose challenges in acquiring sufficient data for optimal performance in certain applications.

In summary, GTR is a transformer-based approach to multi-object tracking that utilizes a global attention mechanism operating on entire frames. It offers several advantages, including the capability to handle occlusions and to track objects across multiple camera views. However, it also has certain limitations, such as its high computational cost and data requirements.

The algorithm is specifically designed for multi-object tracking, a task commonly encountered in various surveillance scenarios, including maritime surveillance. With its impressive features, GTR stands out as a powerful algorithm for multi-object tracking that can be applied to diverse surveillance scenarios, including maritime surveillance. However, it is important to note that deep-learning-based methods such as GTR necessitate a substantial amount of training data and significant computational resources, which can pose challenges in certain applications.

### 4.3. Comparison of the Trackers

This section provides a comparison of the state-of-the-art work in object tracking discussed previously. These studies employ diverse approaches and techniques, such as transformer-based models, Siamese networks, correlation filters, and memory networks. They utilize different evaluation metrics, including precision, recall, and F1-score, and

conduct testing on various datasets such as MOT17 and LaSOT. Through this comparative analysis, we can develop a comprehensive understanding of the strengths and limitations of different object tracking and we can develop a comprehensive understanding of the potential avenues for future advancements in the field.

The studies mentioned in this context employ diverse metrics to assess their tracking performance. However, the selection of metrics can vary depending on the specific task and dataset used for evaluation. Certain studies even introduce novel metrics or modifications to existing ones to effectively evaluate their proposed methods. Consequently, while there are some shared metrics among these studies, the precise choice of metrics may differ based on the research objectives and methodology of each individual work.

In order to establish a link between datasets and metrics, Table 2 summarizes the metrics used in each dataset presented in this section.

**Table 2.** Metrics used in each dataset.

| Dataset | Metrics |
|---|---|
| LaSOT | AUC, $P$ and $P_{norm}$ |
| OTB-2015 | Precision plot, success plot (overlap score, average overlap score, and AUC) |
| VOT-2016 | Accuracy, robustness, and EAO |
| MOT17 | MOTA, IDSW, IDF1, and HOTA |

Table 3 summarizes the performances of the best trackers using all datasets until the writing of this review. The meaning of arrows in Table 3 are as follows: down ↓ means that the lower value is the best and UP ↑ means that the higher value is the best.

**Table 3.** General performance comparison.

| Tracker | AUC | $P_{norm}$ | P | S − AUC | P | A ↑ | R ↓ | EAO ↑ | HOTA ↑ | IDF1 ↑ | MOTA ↑ | $ID_s$ ↓ |
|---|---|---|---|---|---|---|---|---|---|---|---|---|
| 1 | 0.711 | 0.811 | 0.776 | - | - | - | - | - | - | - | - | - |
| 2 | 0.702 | 0.784 | 0.753 | - | - | - | - | - | - | - | - | - |
| 3 | 0.701 | 0.799 | 0.763 | - | - | - | - | - | - | - | - | - |
| 4 | 0.690 | 0.794 | 0.738 | - | - | - | - | - | - | - | - | - |
| 5 | 0.685 | 0.766 | 0.741 | - | - | - | - | - | - | - | - | - |
| 6 | - | - | - | 0.719 | - | - | - | - | - | - | - | - |
| 7 | - | - | - | 0.709 | - | - | - | - | - | - | - | - |
| 8 | - | - | - | 0.712 | - | - | - | - | - | - | - | - |
| 9 | - | - | - | 0.700 | 0.910 | - | - | - | - | - | - | - |
| 10 | - | - | - | 0.692 | 0.922 | - | - | - | - | - | - | - |
| 11 | - | - | - | - | - | 0.677 | 0.224 | 0.466 | - | - | - | - |
| 12 | - | - | - | - | - | 0.564 | - | 0.351 | - | - | - | - |
| 13 | - | - | - | - | - | 0.590 | 0.340 | 0.360 | - | - | - | - |
| 14 | - | - | - | - | - | - | - | - | 65.0 | 80.2 | 80.6 | 1257 |
| 15 | - | - | - | - | - | - | - | - | 64.4 | 79.5 | 79.6 | 1194 |
| 16 | - | - | - | - | - | - | - | - | 61.6 | 76.3 | 75.3 | 1260 |
| 17 | - | - | - | - | - | - | - | - | 59.1 | 71.5 | 75.3 | 2859 |

The studies are numbered as follows: 1—OSTrack [48]; 2—Swintrack [49]; 3—MixFormer [50]; 4—AiAtrack [51]; 5—Unicorn [4]; 6—STMTrack [54]; 7—KeepTrack [55]; 8—KeepTrackFast [55]; 9—AAA [56]; 10—ASRCF Model [57]; 11—SiamMask [3]; 12—SiamVGG [58]; 13—SE-SiamFC [59]; 14—BotSORT [60]; 15—StrongSORT [61]; 16—SimpleTrack [62]; and 17—GTR [63].

## 5. Open Problems and Challenges

The field of object tracking has undergone significant advancements in recent years, with numerous methods and techniques proposed to enhance tracking performance in various scenarios. However, several open problems and challenges still exist, necessitating

further improvements in the accuracy, robustness, and efficiency of object tracking systems. This section explores potential research directions that can advance the state-of-the-art in object tracking. These include the exploration of novel architectures and models; the development of improved object representations and feature extraction methods; the enhancement of data association techniques; and the utilization of additional sources of information, such as contextual cues, semantic segmentation, and object detection, to enhance tracking performance. Additionally, the need for more robust evaluation metrics and benchmarks is highlighted to enable fair comparisons between different tracking algorithms and to facilitate progress in the field.

Despite the advances in the field of object tracking, several open problems still require attention from researchers. Some of the possible open problems are listed next.

- Handling occlusions: It presents a challenge in object tracking as occluded objects are no longer fully visible, making it difficult to maintain accurate identities for each tracked object. Therefore, when multiple objects occlude each other in a video sequence, tracking becomes even more challenging. There is a need to develop robust algorithms that can handle occlusions effectively. In the context of maritime surveillance, occlusions pose a specific challenge as vessels can move behind each other or other obstacles in the scene. This can make it difficult to track the movement of a specific vessel over time.
- Real-time tracking: Real-time tracking is crucial in various applications, including autonomous driving, surveillance, and robotics. While some trackers are designed to operate in real time, many others are not. Achieving real-time performances requires the development of efficient algorithms that can process video frames at high speeds while maintaining accurate tracking. Optimizing computational efficiency, reducing memory requirements, and leveraging hardware acceleration techniques are important considerations for real-time tracking systems. Therefore, there is a need to develop trackers that can process video frames in real time without sacrificing accuracy and increasing complexity.
- Robustness to appearance changes: Objects can undergo variations in illumination, scale, pose, or clothing. This can make it difficult for trackers to recognize the same object across different frames of a video. Developing trackers that can handle appearance changes robustly is a challenging problem. In the context of maritime surveillance, vessel appearance variability adds another challenge, as vessels can differ in size, shape, color, and texture. This variability makes it difficult to identify and track vessels accurately. Furthermore, environmental factors such as lighting conditions, sea state, and weather can significantly impact the performance of tracking systems.
- Robustness to complex scenes: Tracking objects in scenes with multiple moving objects, diverse motion patterns, and varying speeds poses a significant challenge. The presence of numerous objects moving in different directions can hinder trackers in accurately identifying and tracking each individual object. Robustness to complex scenes involves developing algorithms that can handle occlusions, scale changes, cluttered backgrounds, and interactions among objects.
- Generalization to diverse scenarios: Most tracking algorithms are designed to work under specific conditions, such as indoor or outdoor settings, particular camera viewpoints, etc. However, to address the practical requirements of real-world applications, it is essential to develop trackers that can effectively generalize across diverse scenarios.
- Multi-object tracking: Tracking multiple objects in a crowded scene remains a challenging problem. Developing algorithms that can track multiple objects accurately and robustly is an open problem.
- Explainable tracking: It is crucial to understand why a tracker may fail in certain scenarios or how it arrives at specific decisions in order to build trust in the algorithm. Developing explainable tracking algorithms is an open problem.
- Long-term tracking: Most object trackers focus on short-term tracking, i.e., tracking objects over a few seconds. Tracking objects over extended periods is a challenging

problem, especially in large-scale scenes or videos spanning multiple frames, which requires overcoming various difficulties. These challenges include handling appearance changes, variations in pose, and occlusions that may occur over time. Therefore, developing algorithms that can maintain consistent tracking performance over long-term sequences is an ongoing research area.

- Transfer learning: Training object trackers on one dataset does not guarantee their generalization to other datasets. Variations in objects appearances, scales, and other factors across different datasets can hinder the performance of trackers. Developing methods for transfer learning that can improve the generalization ability of object trackers is an important research direction.
- Joint object detection and tracking: Object tracking algorithms should be able to perform joint object detection and tracking, which can improve tracking performance and reduce false positives.
- Robustness to adversarial attacks: Object tracking algorithms should be robust to adversarial attacks, which can be used to fool the algorithm into tracking the wrong object.
- Uncertainty estimation: Object tracking algorithms should be able to estimate uncertainty in the tracking process, which is important for decision-making in applications such as autonomous driving.

These represent only a few of the potential open problems being addressed by researchers in the field of object tracking. There are still unresolved questions regarding the design of effective tracking architectures, the integration of different tracker types, and the most suitable evaluation methodologies for assessing object tracking systems. These areas serve as active domains where researchers are actively investigating novel ideas and methods to enhance object tracking in real-world scenarios. In the following sections, we outline potential directions for future research in the field of object tracking.

While numerous recent studies have investigated the utilization of attention mechanisms in object tracking, there is still ample opportunity for further advancements. Future research could focus on developing more advanced attention mechanisms that can better capture complex relationships between objects and their surroundings context.

Transformer-based models have demonstrated tremendous potential in natural language processing, and recent studies have shown their potential in object tracking as well. Future research could focus on exploring more advanced transformer architectures to improve object tracking performance.

Many existing object tracking algorithms require significant computational resources and can present challenges when attempting real-time execution. Future research could focus on developing more efficient tracking algorithms that can run on low-power devices.

Most existing object tracking algorithms rely primarily on visual information. However, future research could explore the incorporation of additional sensor modalities, such as depth information or light detection and ranging (LIDAR) data, to improve tracking performance in challenging environments.

Several tracking algorithms make the assumption that objects will not leave the field of view or occlude for a long period. However, objects can disappear for extended periods in real-world scenarios, reappear in different locations, or undergo significant appearance changes. Therefore, future research should focus on developing more robust and accurate tracking methods that can handle long-term occlusions and changes in appearance.

The detection of objects on complex backgrounds poses significant challenges in computer vision, including robustness to cluttered scenes, variations in lighting conditions, occlusion, scale, viewpoint, contextual understanding, data imbalance, transferability to real-world scenarios, and interpretability. Achieving accurate detection in the presence of multiple moving objects and complex backgrounds remains an open problem. Lighting changes, occlusion, and variations in scale and viewpoint further complicate the task. Additionally, understanding context, addressing data imbalance, ensuring transferability to real-world environments, and incorporating interpretability are crucial for advancing detection on complex backgrounds. Future research should explore multidisciplinary ap-

proaches, leverage advanced techniques like deep learning and attention mechanisms, foster the availability of diverse datasets, establish standardized evaluation protocols, and focus on addressing these challenges to enhance detection performance in real-world scenarios.

Multi-object tracking remains a challenging task, especially in crowded scenes where multiple objects may occlude each other, or appear and disappear frequently. Future research should focus on developing more effective methods to handle occlusions, re-identifying objects that have disappeared, and accurately associate multiple objects with their corresponding tracks.

Current tracking algorithms are limited in scalability, making applying them in real-time or large-scale scenarios challenging. Future research should focus on developing more scalable tracking methods that can handle large-scale scenarios and run in real time on a standard computer or mobile device.

Recent advances in deep learning have demonstrated significant potential in enhancing tracking performance. Future research can focus on exploring novel deep learning architectures, such as graph neural networks, generative adversarial networks, and attention mechanisms, to enhance tracking accuracy, robustness, and scalability.

Different tracking applications, such as autonomous driving, surveillance, and robotics, possess distinct requirements and challenges. Hence, future research can focus on developing tailored tracking algorithms for specific applications, taking into account the unique characteristics of the application and the specific demands of the task.

In addition to these challenges, there are specific requirements for maritime tracking systems that must be considered. For instance, the system should be able to operate over large distances and handle the movement of vessels in different directions and speeds. Moreover, the system should be able to operate in real time and handle large amounts of data efficiently. These requirements demand high-performance computing and efficient data storage capabilities, which can pose a significant constraint for designing and deploying such systems.

Therefore, developing effective trackers for maritime surveillance requires using advanced computer vision and machine learning techniques, particularly deep-learning-based methods, to handle the complexity and variability of the maritime environment. These techniques can surpass the limitations of traditional image-processing-based methods and can deliver more accurate and robust tracking results. Furthermore, the creation of large-scale datasets tailored to maritime scenarios is crucial for evaluating and comparing the performance of various tracking systems, as well as promoting the development of new approaches.

## 6. Discussion

This review aimed to highlight the treatment of maritime surveillance in the context of object tracking, It provided insights into the metrics employed in recent datasets to assess object tracking performance and how they are diverse. The review also examined notable trackers that utilize various approaches for object tracking and discussed their potential application in maritime tracking scenarios.

### 6.1. Trackers

The algorithms discussed in Section 2 were developed using traditional (i.e., classic) image processing techniques, such as line detection, binarization for segmentation, and subsequent detection for tracking. These methods were developed before the rise of neural network and increase in computational power. One of the main drawbacks of these methods is their sensitivity to environmental changes, as they rely on specific scene characteristics to detect the tracked object. For instance, horizon line detection, background removal, and segmentation using binarization methods were commonly employed. These limitations might justify the lack of comparison of the trackers discussed in that section with other trackers that used newer techniques. While classical methods have shown

satisfactory performance in controlled environments, they may not perform as well in real-world scenarios.

On the other hand, the methods discussed in Section 4 heavily rely on deep learning algorithms, incorporating techniques such as convolutional layers, visual transformers, and neural network models for the detection, recognition, and tracking of objects. These approaches offer the trackers a new level of flexibility, reducing or even eliminating their sensitivity to environmental changes. Employing convolutional layers for feature learning, visual transformers for capturing the object's attention, and neural network models for integrating these techniques into robust tracking systems give the tracker a new level of flexibility.

Convolutional layers play a crucial role in deep-learning-based trackers as they extract relevant features from images. These layers are specifically designed to detect patterns and structures that may be challenging for traditional image processing methods to capture. By doing so, the model learns more abstract representations of the object being tracked, making it more robust to changes in the environment. Visual transformers enable the model to focus its attention on certain parts of the image that are most relevant to tracking an object's movements. This is achieved by learning a set of weights that determine the importance of different parts of the image. By selectively attending to the most relevant parts, the tracker becomes more efficient and accurate. Training the models on extensive datasets of images allows them to learn and recognize objects under a wide range of conditions, including changes in lighting, background, and occlusions. Consequently, the models exhibit adaptability to new environments, displaying robustness in the face of environmental changes and continually improving their performance. Therefore, using deep learning techniques in object tracking has enabled a new level of flexibility and robustness, making these trackers more effective than traditional image-processing-based methods, like the ones used by the works described in Section 2.

*6.2. Datasets*

In Section 4, several datasets were discussed, which are commonly used for evaluating and training object trackers. These datasets encompasses a wide range of scenes, including objects such as cars, motorcycles, persons, bicycles, buses, and boats. The LaSOT dataset, for instance, comprises 70 classes and includes 20 video sequences of boats. Although it may not provide a comprehensive dataset for maritime scenarios, these boat-related video sequences can still be utilized for training and evaluating trackers specifically designed for boat tracking. The boats featured in the LaSOT dataset exhibit diversity, captured from various viewpoints, which is crucial for evaluating a tracker's robustness, considering that the camera is not fixed. Figure 4 shows some examples of video sequences containing boats, displaying the dataset's diversity of boat types, sizes, and orientations.

Among the other datasets presented in this review, LaSOT is the only one that can be used directly for evaluating and training trackers for maritime scenarios since no videos with vessels are found in the other datasets. However, other datasets, such as OTB-2015, VOT2016, and MOT17, are valuable as benchmark datasets for evaluating different tracker proposals, including for SOT and MOT in general. Although these datasets do not include dedicated vessel videos, they contribute to the advancement of object tracking research as a whole and attract researchers to explore the field further.

The utilization of diverse datasets for evaluating and training object trackers plays a crucial role particularly in the development of novel approaches and in evaluating their effectiveness. For example, using benchmark datasets such as OTB-2015, VOT2016, and MOT17, in addition to specialized datasets such as LaSOT, can provide valuable insights for reproducing results in maritime scenarios and beyond.

The absence of dedicated datasets tailored for maritime surveillance poses a challenge when it comes to the development and evaluation of object trackers within this domain. Most datasets available for evaluating object trackers focus on general object tracking, including cars, pedestrians, and bicycles, while neglecting vessels or boats. This lack of

specialized datasets presents difficulties in accurately evaluating the performance of object trackers specifically in maritime scenarios.

Nonetheless, datasets such as LaSOT, containing boat sequences, can offer partial understanding for evaluating object trackers in the maritime domain. However, they do not encompass the complete range of scenarios that can be encountered in maritime surveillance, such as different types of vessels, challenging environmental conditions, and specific tracking requirements. The development of specialized datasets for maritime surveillance can aid in developing and evaluating new tracking approaches and provide insights into the unique challenges encountered within this particular domain.

Specialized datasets can provide a more comprehensive evaluation of tracking algorithms' effectiveness in maritime surveillance, leading to the development of robust and dependable tracking systems. Creating such datasets necessitates substantial effort and resources but can provide long-term benefits for the research community, enabling the development of new techniques that can enhance maritime surveillance applications.

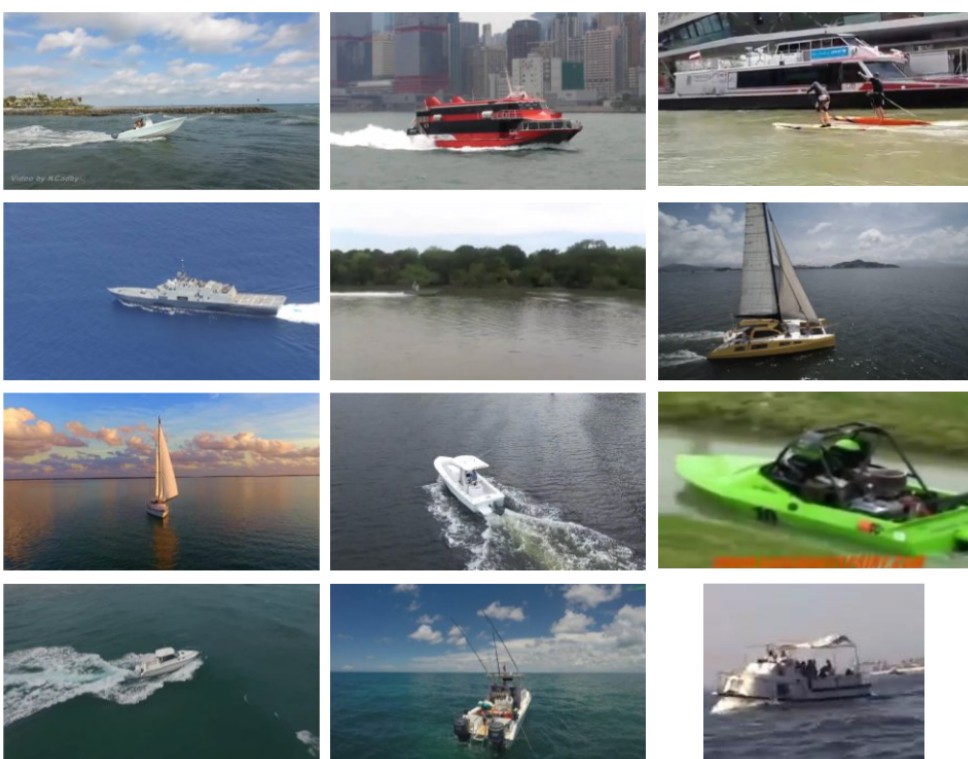

**Figure 4.** Example of frames in video sequences containing boats in LaSOT dataset.

*6.3. Metrics*

The evaluation of tracking algorithms is a crucial task in computer vision, and the metrics used for this evaluation can have a significant influence on the measured performance and effectiveness of these algorithms. However, despite the shared goal of accurately approximating the ground truth bounding box, there is no consensus on the best way to evaluate this performance.

In this review, several metrics were discussed, including precision, normalized precision, accuracy, and expected average overlap (EAO). Precision is commonly used to quantify the distance from the center of the predicted and ground truth bounding boxes. Normalized precision takes into account variations in image sizes by normalizing the distance relative to the diagonal length of the ground truth bounding box. Finally, accuracy computes the overlap between the predicted and ground truth bounding boxes, while EAO extrapolates the frame view to derive an average metric for the entire sequence.

While these metrics share the same objective, they use different formulations. The EAO metric, in particular, offers a comprehensive assessment of tracking by considering the entire video sequence. Trajectory metrics, such as the identity switching metric is used in all the datasets discussed here to measure the tracking robustness in the presence of occlusion.

The EAO and identity switching metrics, used in combination, offer a comprehensive analysis of tracking performance. While the computation of EAO is more complex, it provides valuable insights by considering bounding boxes overlap and provides information about the entire video sequence, including the zero overlaps, also known as failures. On the other hand, identity switching allows the tracker to re-identify an object after an occlusion.

In summary, the lack of unified or common metrics for evaluating object tracking algorithms remains a challenge. As mentioned before, there is no common sense when evaluating a tracking algorithm, and different metrics are used to evaluate the performance of different trackers. The absence of a unified or common set of metrics makes it difficult to compare the performance of different trackers across different datasets and scenarios.

Moreover, the selection of metrics is often dependent on the specific application or scenario under consideration. For instance, a metric that proves effective in tracking small objects in a static scene may not be suitable for tracking larger objects in a dynamic scene. Therefore, it is vital to have a set of metrics available to evaluate trackers in different scenarios but also to have a common baseline that can allow for more direct comparisons. Additionally, the development of new metrics and evaluation protocols should be based on a thorough understanding of the limitations and challenges of object tracking. Metrics solely based on bounding box overlap may fail to capture the intricacies of real-world scenarios, where occlusions, object deformations, and other factors can affect tracking performance.

Hence, it is crucial to strive for the development of a unified or common set of evaluation metrics and protocols in order to advance the field and to facilitate meaningful comparisons between different tracking approaches. However, developing such a set of metrics and evaluation protocols that provide a more accurate and comprehensive assessment of tracking performance across different scenarios can be challenging due to the complexity of tracking tasks and the different evaluation scenarios.

## 7. Conclusions

In conclusion, object tracking has gained significant research interest, leading to the development of numerous datasets and tracking algorithms for various objects and scenarios. However, in the specific context of maritime surveillance, there is a lack of benchmark datasets and tracking algorithms that incorporate the latest advancements in computer vision.

By combining image/video-based tracking technologies with traditional tracking technologies like AIS and radar, it is possible to enhance tracking accuracy in maritime surveillance. However, it is worth noting that tracking in maritime surveillance has predominantly been developed in specific contexts, often utilizing private datasets to address particular challenges, as discussed in Section 2.

However, the discussion presented in this review is insufficient to address the specific requirements of maritime scenarios. To develop tracking systems using the latest techniques, as described in Section 4, a dedicated dataset that encompasses various boat types and diverse scenes is needed. Although vision transformer has been widely used, no specific tracker for ships has been identified. The scarcity of datasets for maritime surveillance hampers the development and evaluation of tracking algorithms in this field, highlighting the need for future contributions from researchers to fill this gap. Furthermore, different metrics have been used in a similar context: object tracking. However, common sense is still lacking regarding trackers since the metrics change with the dataset.

This literature review aimed to provide insights into the development of novel datasets; benchmarking metrics; and primarily, new ship tracking algorithms. However, there is still significant work to be carried out in this field, and this review serves as a foundational starting point for future research in this area.

**Author Contributions:** Conceptualization, R.d.L.R. and F.A.P.d.F.; methodology, R.d.L.R. and F.A.P.d.F.; validation, R.d.L.R. and F.A.P.d.F.; formal analysis, F.A.P.d.F.; writing—original draft preparation, R.d.L.R.; writing—review and editing, F.A.P.d.F.; supervision, F.A.P.d.F.; project administration, F.A.P.d.F.; funding acquisition, F.A.P.d.F. All authors have read and agreed to the published version of the manuscript.

**Funding:** This work was partially supported by the Coordenação de Aperfeiçoamento de Pessoal de Nível Superior-Brazil (CAPES) and by RNP, with resources from MCTIC, grant Nos. 01250.075413/2018-04 and 01245.010604/2020-14, under the 6G Mobile Communications Systems of the Radiocommunication Reference Center (Centro de Referência em Radiocomunicações-CRR) project of the National Institute of Telecommunications (INATEL), Brazil; by Huawei, under the project Advanced Academic Education in Telecommunications Networks and Systems, Grant No. PPA6001BRA23032110257684; by Fundação de Amparo à Pesquisa do Estado de Minas Gerais (FAPEMIG) via Grant No. 2070.01.0004709/2021-28; by FCT/MCTES through national funds and, when applicable, co-funded EU funds under the project UIDB/EEA/50008/2020; by the Brazilian National Council for Research and Development (CNPq) via grant numbers 313036/2020-9 and 403827/2021-3; and by the MCTI/CGI.br and the São Paulo Research Foundation (FAPESP) under grants 2021/06946-0.

**Institutional Review Board Statement:** Not applicable.

**Informed Consent Statement:** Not applicable.

**Data Availability Statement:** Not applicable.

**Conflicts of Interest:** The authors declare no conflict of interest.

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
