# Peer review of "Beyond Land: A Review of Benchmarking Datasets, Algorithms, and Metrics for Visual-Based Ship Tracking"

_electronics, doi:10.3390/electronics12132789_

Round 1

Reviewer 1 Report

The topic of the paper is of scientific importance and the authors are complimented for their efforts. However, this reviewer suggests the authors to make major revisions of the paper and resubmit as it is not recommended for publication in the present form. This reviewer will be pleased to re-review the revised manuscript.
1. Over 47 papers have been reviewed. However, a number of important papers presenting a milestone of maritime surveillance and ship tracking are missing, while articles from the group of authors are included in the survey.
2. Classification of articles should be grouped. And a timeline of all milestones is needed.
3. As far as ship detections are concerned, it needs to clearly note that small ship detection and dynamic ship detection are not included in this paper.
4. The authors are suggested to revisit all of the articles with key words of "Maritime Surveillance and Ship Tracking", published at least in the following journals, although some articles are already included in the paper:
-  IEEE Transactions on Circuits and Systems for Video Technology

-  Expert systems with applications

-  IEEE Transactions on Geoscience and Remote Sensing (
- IEEE Access

5. Certain sections of the content in Section 4 appear to have been automatically generated by an AI system.

Certain sections of the content in Section 4 appear to have been automatically generated by an AI system.

Reviewer 2 Report

The manuscript is very interesting and from a literature review point of view, it tries to adapt the IS and the AI to the shipping industry in order to improve the safety in the maritime traffic. It is very relevant because intends to give a new approach to the use of AIS and radar (with significance to the corresponding manufacturers) as the two principals tools used by sailors to avoiding the collisions between two ships. 

The manuscript address the implementation of benchmark dataset and tracking algorithm to the shipping industry, considering the existing computer vision techniques. This address is considered as relevant because in spite of importance of shipping industry in global economy, the surveillance/control and safety navegation is based on traditional technologies as AIS and radar, which were implemented several decades ago. 

The topic is original and interesting, because it tries to adapt the Intelligent Systems and Artificial Intelligence to the shipping industry with the aim to improve the overall safety in the maritime traffic.  Considering that the vast majority of cargoes is transported by sea; the huge number of different vessels witht different equipment on board of AIS and radar (pleasure ships; fishing vessels, etc.), a revision of state-of-art to improve, for example, the manoeuvers to avoid collisions or monitoring the traffic in restricted sea areas , are considered benefit. 

The start-of-art in this field has been based, for many years, in the use of tradictional AIS and radar systems, both to avoid collisions at sea and to control maritime traffic from the sea shore. Therefore, a new approach trying to put in perspective techniques (as computer vision) used in other fields to this specific sector (shipping industry) is always welcome.  

The main ideas are clearly stated, Conclusions are supported by the review analysis and the discussion carried out, the included references are appropriate and 26 of 47 publications are recent (within the last 5 years), and all Tables and Figures are relevant and are mentioned in the text. The quality of Figures are acceptable too. 

Authors carried out a detailed bibliographical analysis, althoug in some cases, this analysis is considered as excessive. For example, in Section 2 ("Related works"), in my opinion, authors should summarize each of the commented references. 

Furthermore, I consider that manuscript should be considered as a Review not as a Article.  As a review paper, testability, methodological procedures and controls are not applicable. The authors carried out a complete review topic with a manuscript of 38 pages, with relevant and recent publications. Although the goal of manuscript is apply the Intelligent Systems and Artificial Intelligent to maritime industry, a deep analysis of this techniques applied to other different sectors are studied. 

Reviewer 3 Report

This paper reviews current research for object tracking for maritime surveillance and ship tracking.

1. It is recommended that the paper includes up-to-date research with AI to increase the significance. The reviewed research is quite outdated.

2. Please more emphasize the difficulties and limitations in current research.

The quality of English seems to be satisfactory for the future publications. However, the paper can be more improved for better readability.

Round 2

Reviewer 1 Report

All my comments have been fully addressed, and I recommend this paper published